# Can Implicit Bias Explain Generalization?
# Stochastic Convex Optimization as a Case Study

**Assaf Dauber**
Department of Electrical Engineering
Tel-Aviv University
assafdauber@mail.tau.ac.il

**Meir Feder**
Department of Electrical Engineering
Tel-Aviv University
meir@tauex.tau.ac.il

**Tomer Koren**
School of CS, Tel Aviv University
& Google Research Tel Aviv
tkoren@tauex.tau.ac.il

**Roi Livni**
Department of Electrical Engineering
Tel Aviv University
rlivni@tauex.tau.ac.il

## Abstract

The notion of implicit bias, or implicit regularization, has been suggested as a means to explain the surprising generalization ability of modern-days overparameterized learning algorithms. This notion refers to the tendency of the optimization algorithm towards a certain structured solution that often generalizes well. Recently, several papers have studied implicit regularization and were able to identify this phenomenon in various scenarios. We revisit this paradigm in arguably the simplest non-trivial setup, and study the implicit bias of Stochastic Gradient Descent (SGD) in the context of Stochastic Convex Optimization. As a first step, we provide a simple construction that rules out the existence of a *distribution-independent* implicit regularizer that governs the generalization ability of SGD. We then demonstrate a learning problem that rules out a very general class of *distribution-dependent* implicit regularizers from explaining generalization, which includes strongly convex regularizers as well as non-degenerate norm-based regularizations. Certain aspects of our constructions point out to significant difficulties in providing a comprehensive explanation of an algorithm's generalization performance by solely arguing about its implicit regularization properties.

## 1 Introduction

One of the great mysteries of contemporary machine learning is the impressive success of *unregularized* and *overparameterized* learning algorithms. In detail, current machine learning practice is to train models with far more parameters than samples and let the algorithm *fit* the data, oftentimes without any type of regularization. In fact, these algorithms are so overcapacitated that they can even memorize and fit random data. Yet, when trained on real-life data, these algorithms show remarkable performance in generalizing to unseen samples [18, 30].

This phenomenon is often attributed to what is described as the *implicit-regularization* of an algorithm [18]. Implicit regularization roughly refers to the learner's preference to implicitly choosing certain structured solutions *as if* some explicit regularization term appeared in its objective. As a canonical example, in linear optimization one can show that various forms of gradient descent, an apriori unregularized algorithm, behaves identically as regularized risk minimization penalized with the squared Euclidean norm on the parameters [24].

Understanding implicit regularization poses several interesting challenges. For example: how can we find the implicit bias of a given learning algorithm? what is the rate of convergence towards the biased solution? how (and if) does it govern the generalization of an algorithm? and, when and what types of regularizations can account for and explain the generalization in modern-days machine learning?

Towards answering these questions we revisit a fundamental setting that was extensively studied in recent years: Stochastic Convex Optimization (SCO), focusing on the SGD optimization algorithm. In contrast to most previous work, we do not attempt to identify the implicit bias in specific problems. Instead, we study these questions in the general case, and we construct examples which rule out the existence of potential regularizers in general. To some extent, these constructions demonstrate a behavior that might seem counter-intuitive or contradictory to the implicit-bias point of view.

Besides being a well-studied and well-understood model for learning, an important trait of SCO which makes it suitable for our investigation is that learning cannot in general be performed by naive *Empirical Risk Minimization* (ERM). In detail, the work of Shalev-Shwartz et al. [23] showed the existence of SCO instances where naive-ERM fails but *regularized*-ERM succeeds. Thus, we view SCO as a natural test-bed for exploring the role of regularization and its relation to generalization. Compellingly, the generalization of SGD in SCO is well-established, and we are left with the question of how well can we account for generalization through an investigation of its bias.

## 1.1 Contributions

**Implicit distribution-independent bias.** We begin with a simple construction which demonstrates that SGD does not have any *distribution-independent* implicit bias. To show that, we construct a case where SGD *does not* converge to a Pareto-efficient (not even approximately) solution with respect to the empirical loss and a given regularization penalty. In fact, this result is also true for Gradient Descent over smooth functions. In other words, our construction here involves a distribution supported on a single smooth convex function.

Our result is general and rules out any (reasonable) regularizer from being the implicit bias of SGD in this distribution-independent setting. Since the Euclidean-norm distance is the immediate suspect for the implicit regularization of SGD, the first step towards achieving the result is to rule out that Euclidean norm is the implicit bias of SGD. We thus construct an example of a function with a plateau of minimizers where SGD does not converge to the closest point in Euclidean-norm sense. While the result might not seem surprising, it is the technical engine behind the further constructions we provide. Previous to this work, Suggala et al. [27] showed that gradient descent with an infinitely small step size (that is, gradient flow), might diverge from the closest point, and we provide a complementary construction combined with a full rigorous analysis for fixed step-size gradient descent.

**Implicit distribution-dependent bias.** Having ruled out the possibility of a problem-independent regularizer, we proceed to study the more compelling *distribution-dependent* implicit regularization. The question here is whether for every distribution over convex functions, we can associate a regularizer $r$ such that SGD tries to (approximately) find a Pareto-efficient solution with respect to $r$ and the empirical loss (notice that we allow the regularizer to depend on the distribution, but *not* on the specific sample received by SGD.)

We first show that we can rule out the effect of strongly-convex regularizers in the relevant regime of learning (where the dimension and the number of training examples are of roughly the same order). In fact, we rule out a more general class of regularizers that have large range on sets with large diameters. Namely, in any ball with large diameter the regularizer shows preference towards a certain point.

We then continue and demonstrate a distribution where, given an input sample, there is a very large set of possible solutions that share the same empirical loss and the same regularization penalty, and yet, SGD chooses its solution arbitrarily within this set. Here, by "very large" we mean from a learning-theoretic point of view; namely, this set is large enough so that, in general, empirical risk minimization restricted to the set will fail (and yet, it appears that this is exactly what SGD does). In other words, no regularizer $r$ is sufficient for narrowing down the set of possible SGD solutions to the point where non-trivial generalization can be deduced without appealing to other properties of the specific problem.

**Implicit bias in constant dimension.** Several of our constructions are given in high dimension, namely the number of parameters is larger than the number of examples. One could argue that this is the interesting regime, nevertheless it is still worthy to understand the role of implicit bias when the dimension of the problem is smaller than number of examples. Here we cannot rule out the role of implicit bias in a similar fashion to before - namely, due to uniform convergence, any algorithm that is constrained to the unit ball will generalize and this implicit bias is indeed the explanation to that. It is interesting though to understand the existence of specific regularizers (such as, e.g., strongly convex regularizers).

While we do not provide an answer to this question, we make an intermediate step. Our final construction is in a slightly relaxed model, where the instances are non-convex, but the expected loss function is convex. While this result may be limited, because of the non-convexity, we stress that the learning guarantees of SGD are completely applicable to this setting: namely, SGD does learn the problem (as it is convex in expectation). We show that for any *strictly quasi-convex* regularizer, namely a regularizer that has preference for a single point in any convex regime, the algorithm will not converge to the optimal solution with optimal regularization penalty (even though it converges to a convex domain where seemingly it can improve its parameter choice towards the regularized solution).

## 1.2 Related work

Understanding the implicit bias of learning algorithms and its importance in generalization is a central theme in machine learning, and in the study of many classical algorithms [5, 21, 29]. Recently, implicit bias has received considerable attention in the past few years. Starting with [18, 30], it was suggested that implicit regularization might explain the success of networks to improve test error by increasing network size beyond what is needed to achieve zero training error. Subsequently, a line of work has focused on identifying implicit regularization in various problems and domains, e.g., linear and non parametric regression [1, 20, 29] matrix factorization [8, 2], linearly separable data [26, 10], as well as deep networks [17, 19] and others [14, 16, 13, 9]. Our work here can be seen as an attempt to investigate the limitations of implicit regularization. Most similarly to this work, Suggala et al. [27] provides an example of a problem where gradient flow does not converge to closest Euclidean solution. Here we focus on the more concrete SGD algorithm with a fixed step size, and give finite-time analysis. We are also able to harness our example to construct further new constructions that rule out a richer class of implicit-type regularization schemes.

This work can also be seen as an attempt towards separation between *learnability* and *regularization*. Besides regularization, several other useful notions have been suggested as surrogates of learnability. Most classically, uniform convergence [3] has been shown to be equivalent to learnability in the binary, distribution-independent model of PAC learning [28]. As discussed, [23] showed that in the stochastic convex setting naive-ERM fails (but not regularized-ERM), hence learnability and uniform convergence are no longer equivalent. The constructions of Shalev-Shwartz et al. [23] were later substantially strengthened by Feldman [7]. More recently, Nagarajan and Kolter [15] also provided an example that rules out uniform convergence, perhaps in the strictest sense. Their construction, though, does exhibit tangible implicit regularization, which account to the generalization of the algorithm.

Another useful notion is the *stability* of a learning algorithm. Stability is very much related to regularization: e.g., regularizing empirical risk minimization with a strongly convex function induces stability [4], and smoothness can also be harnessed to argue for stability [11]. As such, constructing a convex problem where an algorithm is unstable could also serve as a means to rule out certain types of implicit regularizers. Our examples are in fact stable, and as such, could also be interpreted as a certain weak separation between stability and regularization.

# 2 Preliminaries

## 2.1 The Setup: Stochastic Convex Optimization

We consider the following standard setting of stochastic convex optimization. A learning problem consists of a fixed domain $\mathcal{W}$, which for concreteness we will assume it to be a closed and bounded set in $\mathbb{R}^d$ for some finite $d$, a class of functions $f(\mathbf{w}; z)$ that are convex over $\mathbf{w}$, and an unknown

distribution $D$ over a random variable $z$. The objective of the learner is to minimize:

$$F(\mathbf{w}) := \mathbf{E}_{z \sim D}[f(\mathbf{w}; z)].$$

The goal of the learner, given a sample $S = \{z_1, \ldots, z_T\}$ of $T$ i.i.d. examples from the distribution $D$, is to return a parameter vector $\mathbf{w}_S$ such that

$$\mathbf{E}_S[F(\mathbf{w}_S)] < \min_{\mathbf{w} \in \mathcal{W}} F(\mathbf{w}) + \epsilon, \tag{1}$$

for a desired target accuracy $\epsilon > 0$. (The sample size $T$ may be determined based on $\epsilon$.)

We make the following assumptions throughout. We will generally assume that the functions $f$ are also $O(1)$-Lipschitz. Specifically, in all our constructions we will have $\|\nabla_{\mathbf{w}} f(w, z)\| \le 23$ for all values of $z$ and $\mathbf{w} \in \mathcal{W}$. We will mostly be concerned with the case that $\mathcal{W}$ is a bounded unit ball of radius $r$ around 0. For concreteness we will mostly take $r = 5$. This is just for convenience and clearly our results apply to any constant radius ball. Since our main focus in this paper is on impossibility results, fixing the Lipschitz constant and the diameter does not harm the generality of the setup.

We will also discuss strongly-convex functions (or regularizers): we say that a convex function is $\lambda$-*strongly convex* if for any $\mathbf{w}_1, \mathbf{w}_2 \in \mathcal{W}$ we have: $f(\mathbf{w}_1) \ge f(\mathbf{w}_2) + \nabla f(\mathbf{w}_2)^\top (\mathbf{w}_1 - \mathbf{w}_2) + \lambda \|\mathbf{w}_1 - \mathbf{w}_2\|^2$.

## 2.2 Gradient Descent and Stochastic Gradient Descent

The main focus of this paper is the well-known Stochastic Gradient Descent (SGD) algorithm. Given a sample $S = \{z_1, \ldots, z_T\}$ and a step-size parameter $\eta > 0$, SGD initializes at $\mathbf{w}^{(1)} = \mathbf{0}$ and performs iterations:

$$\forall\, t = 1, \ldots, T: \quad \mathbf{w}^{(t+1)} = \Pi_W\left(\mathbf{w}^{(t)} - \eta \nabla f(\mathbf{w}^{(t)}; z_t)\right), \quad \text{and outputs:} \quad \mathbf{w}_S = \frac{1}{T} \sum_{t=1}^{T} \mathbf{w}^{(t)}, \tag{2}$$

where $\Pi_W(w)$ is defined to be the projection of $w$ over the convex set $W$. The standard SGD analysis guarantees the following (see, e.g., [22]):

**Theorem.** *Let $B, \rho > 0$. Le $\mathcal{W} = \{w : \|w\| \le B\}$, and assume that $F(\cdot)$ is convex and $\|\nabla f(w, z)\| \le \rho$ for all $z$ and $w \in W$. Suppose that SGD is run for $T$ iterations on the sample $S = \{z_1, \ldots, z_T\}$ with step size $\eta = \sqrt{B^2/(\rho^2 T)}$. Then,*

$$\mathbf{E}_S[F(\mathbf{w}_S)] - F(\mathbf{w}^\star) \le \frac{B\rho}{\sqrt{T}}, \tag{3}$$

*where here $\mathbf{w}^\star \in \arg\min_{\mathbf{w}: \|\mathbf{w}\| \le B} F(\mathbf{w})$.*

We will also discuss in this paper the procedure of *Gradient Descent* (GD). Given an objective function $F$ GD obtains the following update steps:

$$\forall\, t = 1, \ldots, T: \quad \mathbf{w}^{(t+1)} = \Pi_W\left(\mathbf{w}^{(t)} - \eta \nabla F(\mathbf{w}^{(t)})\right), \quad \text{and outputs:} \quad \mathbf{w}_F = \frac{1}{T} \sum_{t=1}^{T} \mathbf{w}^{(t)}. \tag{4}$$

In our context, given a sample $S = \{z_1, \ldots, z_T\}$, the gradient descent algorithm takes steps using the full gradient with respect to the empirical loss defined as follows $F_S(\mathbf{w}) = \frac{1}{T} \sum_{t=1}^{T} f(\mathbf{w}, z_t)$. We will then write in shorthand $\mathbf{w}_S$ for $\mathbf{w}_{F_S}$

**Other variants of SGD.** While the above version of SGD is perhaps the most standard one, there are other variants that can be considered. For example, it is common to consider, instead of a fixed step-size, a decaying step-size (where $\eta$ may depend on $t$), as well as taking the last SGD iterate rather than the average iterate. We focus on the version in Eq. (2) for several reasons. First, taking the last iterate is not always justified and attains suboptimal rates (see [25]). Second, the algorithm in Eq. (2) is also the more challenging variant to argue about, in the sense that averaging and taking small fixed step size induces bias towards initialization, and as such, is more strongly regularized (and indeed, the constructions we provide here can be readily modified to address a decaying step-size or the last iterate.[1]) Another variant to consider is *unprojected* gradient descent. Convergence bounds can be

derived for this variant that depend on the norm of the benchmark solution [22, 24]. Again, we note that in all of our constructions we pick domain large enough so that projections in fact don't take place.

Nevertheless, it could be an interesting future work to derive a natural variant of SGD whose implicit regularization properties induce the desired generalization guarantees.

## 2.3 Regularized (Structural) Risk Minimization

Another well studied approach to perform learning is through *regularization*, Regularized Empirical Risk Minimization (ERM) solves the following minimization problem:

$$\widehat{w}_\lambda = \arg \min_{w \in \mathcal{W}} \{F_S(\mathbf{w}) + \lambda r(w)\}, \tag{5}$$

where $\lambda \in \mathbb{R}^+$, and $r(w) : \mathbb{R} \mapsto \mathbb{R}^+$ is a regularization function. When $f(\mathbf{w}; z)$ is Lipschitz-bounded and $r(\mathbf{w}), \lambda$ are properly chosen this method leads to a principled learning algorithm. For example, in the case $r(\mathbf{w}) = \|\mathbf{w}\|^2$, Bousquet and Elisseeff [4] showed that with the correct choice of $\lambda$, Regularized ERM is guaranteed to generalize.

# 3 Regularization

We next discuss the different classes of regularizers we will consider in this paper. While some of the results we provide make little to no assumptions on the regularizers, sometimes we would like to add further structure and rule out specific classes as the implicit bias of SGD, in other cases we would like to formally explain in what sense we might assume that the regularizer does not allow a comprehensive explanation of the implicit bias.

Most generally, a regularizer is any function $r : \mathcal{W} \to \mathbb{R}_+$. We will however make the following basic assumptions on the regularizers, to avoid degenerate cases:

- $\min_{\mathbf{w} \in \mathcal{W}} r(\mathbf{w}) = 0$
- $r$ is non-constant at $\mathcal{W} \backslash \{0\}$;
- $r$ is upper semi-continuous; namely, for every point $\mathbf{w} \in \mathcal{W}$ and every $\epsilon_0 > 0$ there exists a neighborhood $B_{\delta_0}(\mathbf{w}) = \{\mathbf{u} : \|\mathbf{w} - \mathbf{u}\| < \delta_0\}$ for which $r(\mathbf{u}) > r(\mathbf{w}) - \epsilon_0$ if $\mathbf{u} \in B_{\delta_0}(\mathbf{w})$.

Any regularizer that satisfies these properties will be said to be an *admissible regularizer* (or shortly, a regularizer). The first assumption above is only for normalization. For the second assumption, the algorithms we will consider are all initialized at zero and may prefer the zero solution if it is a minimizer of the empirical error. But we are mostly concerned with the implicit bias in more involved cases then that.

The last assumption is perhaps somewhat strongest, but it is intended to rule out pathological examples. For example, one could consider a regularizer $r$ which is 0 on almost all points, but is 1 on the negligible, dense, set of real numbers that SGD would never reach. One could argue that $r$ is an implicit bias of SGD. However, this does not capture our intuition of a regularizer. Thus, we add an assumption that a point penalized by the regularizer should also be penalized under small perturbations.

## 3.1 Strongly-convex Regularizers

While some of the results we will present are given for general (admissible) regularizers, it is natural and expected to study more structured classes of regularizers and ask if they induce the generalization properties of a certain algorithm. One natural family of such regularizers is the class of $\lambda$-*strongly-convex* functions, which we will also assume are 1-Lipschitz. As discussed in length, many of the prominent generalization results are provided in the context of strongly convex regularizers [4, 23].

Strongly-convex regularizers come with a very natural property which allows us to rule out such regularizers on certain problems: a strongly convex function always attains a *unique* minimizer on any convex set. As such we can always identify if the output of an algorithm minimizes (approximately) the strongly-convex regularizer, by comparing the output to the minimizer of the regularizer over the given empirical risk.

## 3.2 General (Admissible) Regularizers

Studying implicit bias that does not stem from a strongly convex regularizer is no less important; however, it becomes much more subtle to rule out the latter. Once the regularizer is allowed to have non-unique minima we should be more careful in stating what we mean when we say it *does not explain generalization*. In fact, almost any plausible algorithm can be said to be implicitly biased on any given distribution. For example, the fact that the regularizer is constrained to the unit ball is a form of algorithmic bias—but as was shown by Shalev-Shwartz et al. [23], it cannot explain generalization in the SCO setting.

Towards clarifying what we mean by "explain generalization", let us consider the following: given a regularizer $r$ and an algorithm $\mathcal{A}$ that outputs a solution $\mathcal{A}(S)$ on a sample $S$, define the set of "competitive" solutions

$$K_{S,r}(\mathcal{A}(S)) = \{\mathbf{w} \in \mathcal{W} \; : \; F_S(\mathbf{w}) \leq F_S(\mathcal{A}(S)) \; \text{ and } \; r(\mathbf{w}) \leq r(\mathcal{A}(S))\}. \tag{6}$$

For shorthand, we will also use the notation $K_{S,r}(\mathcal{A})$ instead of $K_{S,r}(\mathcal{A}(S))$.

In words, $K_{S,r}(\mathcal{A})$ is the set of solutions that are comparable with (or better than) the output of $\mathcal{A}$, with respect to both the empirical loss and the regularization penalty. For example, consider a regularized ERM, as in Eq. (5), then $K_{S,r}(\mathcal{A})$ depicts *all* minimizers of Eq. (5) with comparable regularization penalty. For example, with a strongly-convex regularizer $r$ one can observe that the set $K_{S,r}(\mathcal{A})$ is in fact a set of a single *unique* solution.

More generally, if a regularizer $r$ is said to be the implicit bias of an algorithm $\mathcal{A}$, and as such it explains the generalization of the algorithm, it is expected that the set $K_{S,r}(\mathcal{A})$ would be "small" in the sense that choosing an arbitrary solution from it should provide principled guarantees. If we cannot attain such guarantees without further investigation of the problem and algorithm, we argue that the regularizer does not provide a comprehensive explanation of generalization. This motivates the following definition for studying more general regularizers than, say, strongly convex ones:

**Definition 1.** *Let us say that a set $K$ is $(T, \epsilon_0)$-statistically complex if for some distribution $D$ over 1-Lipschitz convex functions, given $T$ i.i.d. samples we have that with probability at least $1/10$ that for some $\mathbf{w} \in K$ it holds that $\frac{1}{T} \sum_{i=1}^{T} f(\mathbf{w}, z_i) = 0$, yet $\mathbf{E}_z[f(\mathbf{w}, z)] > \epsilon_0$.*

Note that the statistical complexity of the set $K$ is measured with respect to an *arbitrary* distribution $D$ over convex functions: this captures our requirement that the set $K_{S,r}(\mathcal{A})$ should explain generalization, *without further investigation of the problem*. In other words, it could be that for a correct choice of a regularizer, on a specific problem, all the models in $K_{S,r}(\mathcal{A})$ will generalize. However, what we want is to ensure that the generalization does not stem from any further structure in the problem that is not captured by the regularizer. Thus, we require that this set will be "simple" in the sense that on any arbitrary distribution over convex functions we can choose an arbitrary solution that minimizes the empirical risk.

## 4 Results

### 4.1 Distribution Independent Implicit Regularization

We start with the natural question, whether there is some distribution independent implicit regularization being promoted by SGD. As a warm-up we begin by ruling out the existence of a distribution-independent strongly convex regularizer that plays the role of the implicit bias of SGD. This family of regularizers is already very interesting, and has been studied extensively in the literature of stochastic convex optimization [4, 23].

**Theorem 1.** *Let $\mathcal{W} = \{\mathbf{w} : \|\mathbf{w}\| \leq 5\}$. For every 1-Lipschitz and $\lambda$-strongly convex $r$, there is a distribution $D_r$ over 1-Lipschitz and 1-smooth functions over $\mathcal{W}$, and $\mathbf{w}_r \in \mathcal{W}$ such that, with probability 1, SGD with any step size $1/T^2 < \eta < 1$ over an input sample $S$ of size $T = \Omega(1/(\lambda\eta))$ outputs $\mathbf{w}_S$ such that:*

$$F_S(\mathbf{w}_r) \leq F_S(\mathbf{w}_S), \quad \text{and} \quad r(\mathbf{w}_r) \leq r(\mathbf{w}_S) - \Theta(\lambda) .$$

In words, for any strongly convex regularizer there exists an instance problem where SGD chooses a solution that is sub-optimal in terms of both empirical error, and regularization penalty.

The last result can be extended to general (admissible) regularizers. Here, the rate of divergence from a Pareto optimal solution depends on the structure of the regularizer $r$. This dependence of the divergence-rate on the regularizer $r$ is unavoidable. Indeed if we consider a regularizer $r$ such that $r \approx 0$, it is not hard to be convinced that it would take SGD longer to become $r$-suboptimal.

**Theorem 2.** *Let $\mathcal{W} = \{\mathbf{w} : \|\mathbf{w}\| \leq 5\}$. For every admissible regularizer $r$, there are constants $c_r > 0$, a distribution $D_r$ (over 1-Lipschitz and 1-smooth convex functions), and $\mathbf{w}_r \in \mathcal{W}$ such that, with probability 1 over the input sample S, SGD with any step size $1/T^2 < \eta < 1$ and sample size $T_r = \Omega_r(1/\eta)$ outputs $\mathbf{w}_S$ such that:*

$$F_S(\mathbf{w}_r) \leq F_S(\mathbf{w}_S), \quad and \quad r(\mathbf{w}_r) \leq r(\mathbf{w}_S) - c_r \ .$$

The $\Omega_r(\cdot)$ notation hides constant that may depend on the regularizer $r$. The dependence on the regularizer is expected here, as we would need a very strong level of accuracy if we want to rule out a nearly-constant regularizer, for example.

## 4.2 Distribution-Dependent Implicit Regularization

Having ruled out a class of implicit regularizers in the distribution-independent model, we next move on to discuss the possibility of distribution dependent regularizers.

**Theorem 3.** *For every $T \geq 1$, a constant $C > 2$ and dimension $d > T/10$: there exists a distribution $D$ over 1-Lipschitz convex functions over $\mathcal{W} = \{\mathbf{w} : \|\mathbf{w}\| \leq 1\} \subseteq \mathbb{R}^d$, such that if we run SGD with learning rate $1/T^2 < \eta \leq C/\sqrt{T}$ over a sample set of size T, then for any 1-Lipschitz, $\lambda$-strongly convex regularizer $r$, with probability 0.1 over the sample, SGD outputs $\mathbf{w}_S$ for which there is $\mathbf{w}^\star \in \mathcal{W}$, such that*

$$F_S(\mathbf{w}^\star) \leq F_S(\mathbf{w}_S), \quad and \quad r(\mathbf{w}^*) \leq r(\mathbf{w}_S) - 10^{-2}\frac{\lambda T \eta^2}{C} \ .$$

Utilizing a construction of a statistically complex set due to Feldman [7], we can also obtain the following result:

**Theorem 4.** *For every $T \geq 1$, a constant $C > 2$ and dimension $d \geq T/10^5$: there exists a distribution $D$ over convex 1-Lipschitz functions over $\mathcal{W} = \{\mathbf{w} : \|\mathbf{w}\| \leq 1\} \subseteq \mathbb{R}^d$, such that if we run SGD with stepsize $1/T^2 < \eta \leq C/\sqrt{T}$ over a sample set of size T, then for any regularizer $r$ we have that with probability at least $1/10$ over the sample, the set $K_{S,r}(\mathbf{w}_S)$ is $\left(2T, 10^{-5}\frac{T\eta^2}{C}\right)$-statistically complex.*

In words, Theorem 4 asserts that for a certain given distribution $D$ the output of SGD cannot be interpreted as coming from a "small" structured family of solutions that would generalize regardless of other specialized properties of the particular learning problem.

The requirement that $T \leq O(d)$ is tight. Note that for a sample $S$ of order $T = O(d/\epsilon^2)$, by a standard covering argument, we can show that the set $K_{S,r}(\mathcal{W})$ is not $(2T, \epsilon)$ statistically complex (see, for example, Theorem 5 of [23]). In particular, since $K_{S,r}(\mathbf{w}_S) \subseteq \mathcal{K}_{S,r}(\mathcal{W})$ we obtain an upper bound of the statistical complexity of the given set.

## 4.3 Implicit Bias in Constant Dimension

In the results above we provided constructions in spaces with more parameters than samples. We next discuss the case $d \ll T$, which is interesting for certain contexts.

Regarding Theorem 4, we again point out that such a result cannot hold in the aforementioned regime. Indeed, in this case uniform convergence over the unit-ball applies. In that sense, restricting an algorithm to choose a solution in the unit ball provides an inductive bias that provides generalization guarantees. But what about Theorem 3? It is interesting to know if one can rule out regularizers that are not benign like the unit ball.[2] We do not know the answer to this question and we leave it as an open problem. Nevertheless, we can provide the following intermediate result in a slightly more relaxed setting, where the instances may be non-convex, (and in fact non-Lipschitzian) but the

expected loss function is indeed convex, and at each iteration the learner observes a bounded gradient $\|\nabla f(\mathbf{w}, z)\| \leq 1$ Thus, SGD's learning guarantee still apply.

We will state the next result for a slightly larger class of regularizers than merely convex regularizers. Recall that a function $f$ is called *quasi-convex* if $f(\lambda x + (1 - \lambda)y) \leq \max\{f(x), f(y)\}$ for every $0 \leq \lambda \leq 1$ and $x, y \in \mathcal{K}$, and *strictly* quasi-convex if $f(\lambda x + (1 - \lambda)y) < \max\{f(x), f(y)\}$.

**Theorem 5.** *Let $\mathcal{W} = \mathbb{R}^2$. There exists a distribution $D$ over, not necessarily convex, functions in $\mathbb{R}^2$ such that $\mathbf{E}_z[f(\mathbf{w}; z)] = 0$ for every $\mathbf{w} \in \mathcal{W}$, and for every strictly quasi-convex regularizer $r$, and for large enough $T$, if $\eta = \Theta(1/\sqrt{T})$ then with some positive probability, $\Theta(1)$, there exists $\mathbf{w}^\star$ such that:*

$$F_S(\mathbf{w}^\star) \leq F_S(\mathbf{w}_S); \qquad r(\mathbf{w}^\star) < r(\mathbf{w}_S); \qquad \|\mathbf{w}_S - \mathbf{w}^\star\| = \Theta(1).$$

## 5 Constructions

Here we give a high level description of our first construction which rules out any norm-based distribution-independent regularizers and any strongly convex distribution-independent regularizer (Theorem 1). This construction is also the basis of the rest of the results (Theorems 2 to 5). The other constructions, as well as the full proofs, are provided in the full-version [6]. We note that for simplicity of exposition, the following description refers to the last iterate, but our full proofs refers to Eq. (2) (i.e., the algorithm that outputs $\mathbf{w}_S = \frac{1}{T} \sum_{t=1}^{T} \mathbf{w}^{(t)}$) .

### 5.1 Distribution Independent Regularization

Our constructions build upon the following class of functions in $\mathbb{R}^2$. Let $A$ be a set of the form $\{(\alpha, \theta) : 0 \leq \alpha \leq b\}$, where $\theta, b$ are parameters of the set and $\Sigma$ is a PSD matrix. We then consider the function $f_{A,\Sigma}$ defined as follows:

$$f_{A,\Sigma}(\mathbf{w}) = \frac{1}{2} \min_{\mathbf{v} \in A} \left\{ (\mathbf{w} - \mathbf{v})^\mathsf{T} \Sigma (\mathbf{w} - \mathbf{v}) \right\}. \tag{7}$$

One can observe that these functions are convex, and further the gradient of $f_{A,\Sigma}$ at point $\mathbf{w}$ will equal

$$\nabla f_{A,\Sigma}(\mathbf{w}) = \Sigma(\mathbf{w} - \mathbf{v}(\mathbf{w})), \qquad \text{where} \qquad \mathbf{v}(\mathbf{w}) = \arg\min_{\mathbf{v} \in A} \{(\mathbf{w} - \mathbf{v})^\mathsf{T} \Sigma (\mathbf{w} - \mathbf{v})\}. \tag{8}$$

**Warm-up: GD need not converge to a minimal-norm solution.** We start by showing how we can construct a function (of the type in Eq. (7)) that does not converge to minimal norm solution. Let us take a concrete case where

$$A = \{(\alpha, 1) : 0 \leq \alpha \leq \infty\}, \qquad \Sigma = \begin{pmatrix} 1 & \frac{1}{2} \\ \frac{1}{2} & 1 \end{pmatrix}.$$

We will suppress dependence on $A$ and $\Sigma$, and simply write $f$. The main observation is that the trajectory of $f$ is characterized by two phases.

At the first phase the closest point to $\mathbf{w}^{(t)}$ (with respect to the $\Sigma$-norm) is at the boundary of $A$ (i.e $\alpha = 0$). At this phase, $\mathbf{w}^{(t)}$ can be seen to move "towards" the center of the interval, namely $w_1^{(t)}$ is increasing (see Eq. (8)). At the end of this phase, $w_1^{(t)}$, is sufficiently large irrespective of the step size $\eta > 0$. The second phase, starts when $\mathbf{e}_2 \equiv \binom{0}{1}$ stops being the closest point, and the closest point to $\mathbf{w}^{(t)}$ is at the interior of the interval. One can show that at this phase, the gradient moves upward hence $w_1^{(t)}$ does not decrease and overall the trajectory will converge to a point away from $\mathbf{e}_2$: the Euclidean closest minimizer to 0.

To see that when $\mathbf{v}(\mathbf{w})$ is at the interior of $A$ then $\nabla f(\mathbf{w}) \propto \mathbf{e}_1$, consider the following scalar function $g(a) = (\mathbf{w} - (a, 1))^\mathsf{T} \Sigma (\mathbf{w} - (a, 1))$. Our assumption is that $g$ attains its minimum at $0 < v_1$. Taking the derivative at $v_1$ and equating to 0 (because the minimum is attained at the interior), we can see that $g'(v_1) = (\mathbf{w} - (v_1, 1))^\mathsf{T} \Sigma \mathbf{e}_1 = 0$. Hence, $\nabla f(\mathbf{w}) = (\mathbf{w} - v(\mathbf{w}))\Sigma \perp \mathbf{e}_1$. We depicted here the trajectory of GD without the projection step, however one can observe that throughout, the algorithm never escapes the 2-ball, hence projections are indeed never implemented. The trajectory of $\mathbf{w}^{(t)}$ is illustrated in **??** (green line).

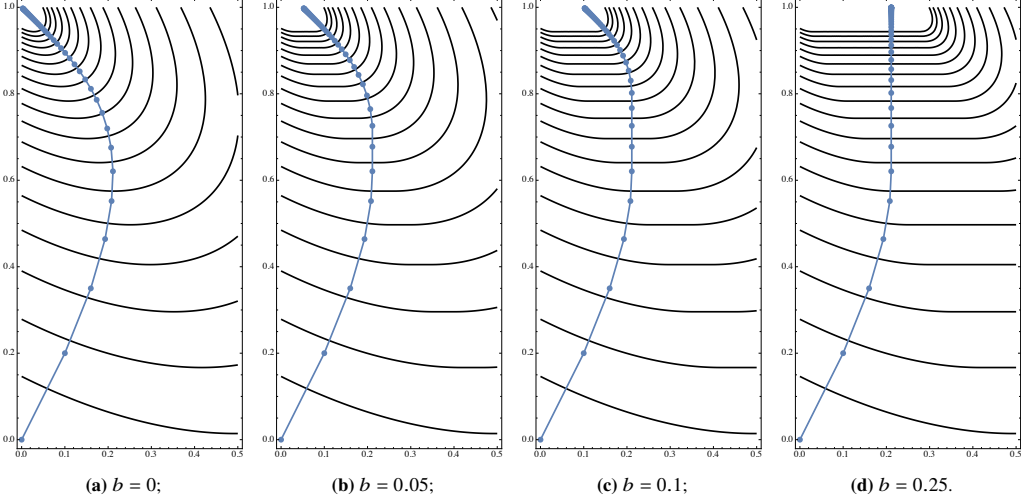

| (a) $b = 0$; | (b) $b = 0.05$; | (c) $b = 0.1$; | (d) $b = 0.25$. |

**Figure 1:** Simulation of GD (with step size $\eta = 0.2$) on $f_{A,\Sigma}$ for $\theta = 1$ and varying values of $b$. We see that GD does not necessarily converge to the nearest solution, and tuning $b$ changes the point towards which it is biased.

**No strongly-convex implicit bias (Theorem 1).** The construction above is the heart of most of our results. Let us illustrate how it rules out a strongly convex regularizer (in the distribution-independent setting) and attain Theorem 1.

The key property of strongly-convex regularizers is that in any convex set they have a unique minimum. Moreover, two far away points cannot simultaneously attain close-to-minimal value. This is in fact the only property we will use. Thus, our result can in fact be extended to any regularizer that is a "tie-breaker"—namely, it always prefers a single unique solution amongst a class of possible solutions with large diameter.

The construction above will allow us to generate two instances of convex learning problems, where SGD converges to two far away points. The first instance is the standard Euclidean distance. Namely, we take a function $f_1$ of the form in Eq. (7), with $\Sigma$ the identity and $A$ with boundaries $(-\infty, \infty)$. In this case, SGD is biased towards the nearest solution $\mathbf{e}_2 = (0, 1)$. The second instance, $f_2$, is the construction above where SGD is biased towards another point on the interval (see Figs. 1b to 1d).

Now both points are global minima, for both $f_1$ and $f_2$, hence if SGD is implicitly biased towards solutions with minimum regularization penalty $r$, we must have that $r(\mathbf{e}_2) = r(\mathbf{v})$, where $\mathbf{v}$ is the choice of SGD when it observes $f_2$. However, if $r$ is strongly convex, because $\|\mathbf{e}_2 - \mathbf{v}\| = \Theta(1)$, there has to be a point on the interval between them that attain a strictly lesser regularization penalty, moreover it also attains minimal loss value. This contradicts the existence of such an $r$.

**The general case.** Our second result (Theorem 2) rules out the existence of any distribution-independent regularizer. In contrast with the strongly-convex case we can not give uniform bounds that depend on parameters of strong convexity. As such, the rates depend on the regularizer.

But the construction here is similar. We basically start with the assumption that there are two points $\mathbf{w}_1$ and $\mathbf{w}_2$ with different regularization penalty, and we want to construct two functions $f_1, f_2$ that maps $\mathbf{w}_1, \mathbf{w}_2$ to the same empirical loss. It might seem that through a simple linear transformation that maps, say, $\mathbf{w}_1$ to $\mathbf{e}_2$ and $\mathbf{w}_2$ to $\mathbf{v}$ we can reduce this case to the case above. However, there is some subtlety since gradient descent is not invariant to linear transformations.[3]

Towards this, we extend the construction above by constructing a more general example, where we can tune the point of convergence of SGD to *any* point on the interval between $\mathbf{v}$ and $\mathbf{e}_2$. This allows us to avoid scaling, and use only rotations (which SGD is invariant to) in order to reduce the problem to the former case. This is done by changing the set $A$ from allowing $0 \le \alpha < \infty$ and $\theta = 1$, to adding a second boundary condition on the right and also scaling $\theta$. In Fig. 1 we illustrate how changing the boundary condition changes the trajectory.

## Broader Impact

There are no foreseen ethical or societal consequences for the research presented herein.

## Acknowledgments and Disclosure of Funding

TK is supported by an ISF grant no. 2549/19 and by the Yandex Initiative in Machine Learning. RL is supported by an ISF grant no. 2188/20 and partially funded by an unrestricted gift from Google. Any opinions, findings, and conclusions or recommendations expressed in this work are those of the author(s) and do not necessarily reflect the views of Google. MF is supported by an ISF gran no. 819/20.

## Footnotes

[1]In fact, the proofs will be significantly simpler; for example, in the proof overview we actually consider the last iterate for simplicity.

[2]We treat a set $K$ as a regularizer by identifying $K$ with a regularizer $r$ such that $r(\mathbf{w}) = 1$ if $\mathbf{w} \notin K$ and 0 otherwise.

[3]We note though that it can be turned to an affine invariant optimization algorithm [12].

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
