[Supplementary Material]

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

We do not know the answer to this question and we leave it as an open problem. Nevertheless we can provide the following intermediate result in a slightly more relaxed setting, where the instances may be non-convex, but the expected loss function is indeed convex. Thus, SGD's learning guarantee still implies.

We will state the next result for a slightly larger class of regularizers than merely convex regularizers. Recall that a function $f$ is called *quasi-convex* if $f(\lambda x + (1 - \lambda)y) \leq \max\{f(x), f(y)\}$ for every $0 \leq \lambda \leq 1$ and $x, y \in \mathcal{K}$. and *strictly* quasi-convex, if $f(\lambda x + (1 - \lambda)y) < \max\{f(x), f(y)\}$

**Theorem 5.** *There exists a distribution $D$ over $\mathbb{R}^2$, on not necessarily convex functions, such that $\mathbb{E}_z[f(\mathbf{w}; z)] = 0$ for every $\mathbf{w} \in \mathcal{W}$, for every strictly quasi-convex regularizer $r$, and for large enough $T$, if $\eta = \Theta(1/\sqrt{T})$ then with some positive probability, $\Theta(1)$, there exists $\mathbf{w}^\star$ such that:*

$$F_S(\mathbf{w}^\star) \leq F_S(\mathbf{w}_S); \qquad r(\mathbf{w}^\star) < r(\mathbf{w}_S); \qquad \|\mathbf{w}_S - \mathbf{w}^\star\| = \Theta(1).$$

# 5 Constructions

Here we give a high level description of the constructions as well as the proofs of the main results. We remark that our full proofs address the average of the SGD iterates (as presented in Eq. (2)); for simplicity of exposition, though, here we will mainly focus on the last iterate.

## 5.1 Distribution Independent Regularization

Our constructions build upon the following class of functions in $\mathbb{R}^2$. Let $A$ be a set of the form $\{(\alpha, \theta) : b_1 \leq \alpha \leq b_2\}$, where $\theta, b_1, b_2$ are parameters of the set and $\Sigma$ is a PSD matrix. We then consider the function $f_{A,\Sigma}$ defined as follows:

$$f_{A,\Sigma}(\mathbf{w}) = \frac{1}{2} \min_{\mathbf{v} \in A}\{(\mathbf{w} - \mathbf{v})^\top \Sigma (\mathbf{w} - \mathbf{v})\}. \tag{7}$$

One can observe that these functions are convex, and further the gradient of $f_{A,\Sigma}$ at point $\mathbf{w}$ will equal

$$\nabla f_{A,\Sigma}(\mathbf{w}) = \Sigma(\mathbf{w} - \mathbf{v}(\mathbf{w})), \tag{8}$$

where

$$\mathbf{v}(\mathbf{w}) = \arg\min_{\mathbf{v} \in A}\{(\mathbf{w} - \mathbf{v})^\top \Sigma (\mathbf{w} - \mathbf{v})\}.$$

**Warm-up: GD need not converge to a minimal-norm solution.** We start by showing how we can construct a function (of the type in Eq. (7)) that does not converge to minimal norm solution. Let us take a concrete case where

$$A = \{(\alpha, 1) : 0 \leq \alpha \leq \infty\}; \qquad \Sigma = \begin{bmatrix} 1 & 1/2 \\ 1/2 & 1 \end{bmatrix}.$$

We will suppress dependence on $A$ and $\Sigma$, and simply write $f$. The main observation is that the trajectory of $f$ is characterized by two phases.

At the first phase the closest point to $\mathbf{w}^{(t)}$ (w.r.t. $\Sigma$-norm) is at the boundary of $A$ (i.e $\alpha = 0$). At this phase, $\mathbf{w}^{(t)}$ can be seen to move "towards" the center of the interval, namely $w_1^{(t)}$ is increasing (see Eq. (8)). At the end of this phase, $w_1^{(t)}$, is sufficiently large irrespective of the step size $\eta > 0$.

The second phase, starts when $\mathbf{e}_2 \equiv \begin{pmatrix} 0 \\ 1 \end{pmatrix}$ stops being the closest point, and the closest point to $\mathbf{w}^{(t)}$ is at the interior of the interval. One can show that at this phase, the gradient moves upward hence $w_1^{(t)}$

does not decrease and overall the trajectory will converge to a point away from $\mathbf{e}_2$: the Euclidean closest minimizer to 0.

To see that when $\mathbf{v}(\mathbf{w})$ is at the interior of $A$ then $\nabla f(\mathbf{w}) \propto \mathbf{e}_1$, consider the following scalar function:

$$g(a) = (\mathbf{w} - (a,1))^T \Sigma (\mathbf{w} - (a,1)).$$

Our assumption is that $g$ attains its minimum at $0 < v_1$. Taking the derivative at $v_1$ and equating to 0 (because the minimum is attained at the interior), we can see that $g'(v_1) = (\mathbf{w} - (v_1,1))^T \Sigma \mathbf{e}_1 = 0$. Hence, $\nabla f(\mathbf{w}) = (\mathbf{w} - v(\mathbf{w}))\Sigma \perp \mathbf{e}_1$.

The trajectory of $\mathbf{w}^{(t)}$ is illustrated in Fig. 1 (green line).

**No strongly-convex implicit bias.** The construction above is the heart of most of our results. Let us illustrate how it rules out a strongly convex regularizer (in the distribution-independent setting) and attain Theorem 1.

The key property of strongly-convex regularizers is that in any convex set they have a unique minimum. Moreover, two far away points cannot simultaneously attain close-to-minimal value. This is in fact the only property we will use. Thus, our result can in fact be extended to any regularizer that is a "tie-breaker"- namely it always prefers a single unique solution amongst a class of possible solutions with large diameter.

The construction above will allow us to generate two instances of convex learning problems, where SGD converges to two far away points. The first instance is the standard Euclidean distance. Namely we take a function $f_1$ of the form in Eq. (7), with $\Sigma$ the identity and $A$ with boundaries $(-\infty, \infty)$. In this case SGD is biased towards $\mathbf{e}_2$ (gray dashed line in Fig. 1). The second instance, $f_2$, is the

Figure 1: Simulation of objective functions where GD does not converge to closest point. At the first phase, the gradient points toward right: specifically the vector $\Sigma \mathbf{e}_2$ (see Eq. (7)), which causes it to diverge from $\mathbf{e}_2$. The red, orange, blue and green trajectories are simulations of the same objective function when we replace $A$ from *Eq.* (7) with $A' = \{(\alpha, 1), 0 \leq \alpha \leq b\}$ for values $b = \{0, 0.1, 0.15, \infty\}$ respectively.

construction above where SGD is biased towards another point on the interval (green line in Fig. 1).

Now both points are global minima, for both $f_1$ and $f_2$, hence if SGD is implicitly biased towards solutions with minimum regularization penalty $r$, we must have that $r(\mathbf{e}_2) = r(\mathbf{v})$, where $\mathbf{v}$ is the choice of SGD when it observes $f_2$. However, if $r$ is strongly convex, because $\|\mathbf{e}_2 - \mathbf{v}\| = O(1)$, there has to be a point on the interval between them that attain a strictly lesser regularization penalty, moreover it also attains minimal loss value. This contradicts the existence of such an $r$.

**The general case.** Our second result (Theorem 2) rules out the existence of any distribution-independent regularizer. In contrast with the strongly-convex case we can not give uniform bounds that depend on parameters of strong convexity. As such, the rates depend on the regularizer.

But the construction here is similar. We basically start with the assumption that there are two points $\mathbf{w}_1$ and $\mathbf{w}_2$ with different regularization penalty, and we want to construct two functions $f_1, f_2$ that maps $\mathbf{w}_1, \mathbf{w}_2$ to the same empirical loss. It might seem that through a simple linear transformation that maps, say, $\mathbf{w}_1$ to $\mathbf{e}_2$ and $\mathbf{w}_2$ to $\mathbf{v}$ we can reduce this case to the case above. However, there is some subtlety since gradient descent is not invariant to linear transformations.[3]

Towards this, we extend the construction above by constructing a more general example, where we can tune the point of convergence of SGD to *any* point on the interval between $\mathbf{v}$ and $\mathbf{e}_2$. This allows us to avoid scaling, and use only rotations (which GD is invariant to) in order to reduce the problem to the former case. This is done by changing the set $A$ from allowing $0 \leq \alpha < \infty$, to adding a second boundary condition on the right. In Fig. 1 we illustrate how this changes the trajectory (red, orange and blue lines).

## 5.2 Distribution-Dependent Implicit Bias

We next discuss our second sets of results that argue about *distribution-dependent regularization*. Here we want to study if, for a given distribution, the set of solutions on which SGD converges has some

Figure 2: Depiction of the auxiliary construction in Lemma 1. The left sketch illustrate the first function where the gradient of the loss always points upward, hence on a $x$-axis parallel line it is constant. The right figure illustrate the second loss function. Here at a regime that include 0 the gradient points sideways, and at the second regime, the gradient points vertically upwards

meaningful structure on which we can argue why it generalizes.

Note that so far, our problem instances considered only a single function and the results were applicable to GD also. Here, though, in the distribution dependent setting such an example cannot work. Indeed, given a single function as an instance problem, SGD behaves deterministically and the solution it chooses is a unique solution which trivially generalizes.

**No strongly-convex distribution-dependent bias.** We next discuss our argument that rules out a strongly convex regularizer, even if it may depend on the distribution at hand. We again utilize the property that a strongly convex regularizer obtains approximately minimum solutions only on a small diameter around the unique minimum.

Our strategy is as follows: assume that there are two samples $S_1$ and $S_2$ such that, when SGD observes $S_1$ it converges to $\mathbf{w}_1$ and when it observes $S_2$ it converges to $\mathbf{w}_2$. However, assume also that $\|\mathbf{w}_1 - \mathbf{w}_2\| = \Theta(1)$, and that the empirical loss of $\mathbf{w}_1$ and $\mathbf{w}_2$ is comparable, on both samples: namely $F_{S_1}(\mathbf{w}_1) = F_{S_2}(\mathbf{w}_1)$, and similarly with $\mathbf{w}_2$.

In the case above, as we argued in the distribution-independent case, clearly the algorithm failed to choose the minimizer of the regularization penalty, in at least one of the realization $S_1$ or $S_2$. So if $S_1$ and $S_2$ are equally likely, we obtain that with probability half (conditioned on the event that we saw $S_1$ or $S_2$) the algorithm failed to minimize $r$. Now, if the probability to observe one of such couple of samples $S_1, S_2$ is positive, then we obtain the desired result.

To generate this setting, we rely on the following auxiliary construction in $\mathbb{R}^2$. We construct two functions such that, if SGD observes the first function, at the first iteration, then the gradient points upward and right. But if SGD observes the second function, at the first iteration, then the gradient points upward. This ensures that in each case SGD will move towards a different solution. If the size of the gradient is constant then the gap between the two iterations will be $\Theta(\eta)$.

We will also construct the examples in such a way that both points enter a regime where all points obtain the same empirical loss on both functions. This construction can in fact be done using piece-wise linear functions and it is illustrated in Fig. 2. We also give the formal statement here:

**Lemma 1.** *For every constant $0 < c < 1$, there are two $1$-Lipschitz functions $f(\mathbf{w}; \pm 1)$ over $\mathbb{R}^2$ such that if $\mathbf{v}_1 = -\nabla f(0; 1)$ and $\mathbf{v}_{-1} = -\nabla f(0; -1)$ and $c < \frac{1}{2}\eta < 1$ then $\|\mathbf{v}_1 - \mathbf{v}_{-1}\| \geq 1/4$ and $f(\eta \mathbf{v}_1; z) = f(\eta \mathbf{v}_{-1}; z)$ for any $z \in \{-1, 1\}$.*

We next utilize the above construction to generate the problem in $\mathbb{R}^d$. Note that the construction above generates a problem where SGD will converge to two different solutions with distance $\eta$ but same empirical loss (after one step). Indeed, we just need to randomly pick one of these functions.

We next want to amplify the distance. To do that, we consider $d = \Omega(T)$ Cartesian copies of $\mathbb{R}^2$. Then at each example, we show one of the functions above, at one of the products. Assuming enough coordinates were seen only once (which is going to happen w.h.p.), the variance on each sub-plane will be $\eta^2$: if we have $\Theta(T)$ such coordinates, the overall variance is going to be $\Theta(T\eta^2)$ which ensures that we will converge to far away solutions on different realizations of the problem, if $\eta = \Theta(1/\sqrt{T})$.

**SGD might be biased towards statistically-complex sets.** Next, we derive Theorem 4 which addresses implicit regularization in a much broader setting. As discussed, here we cannot rule out the existence of an implicit bias; indeed, some form of an implicit bias always exists. We attempt, though, to understand how the implicit bias can explain generalization.

The result shows that for any regularizer: the set $K_{S,r}(\mathbf{w}_S)$ which is the set of comparable solutions to the one outputted by SGD, given the empirical loss and regularization penalty, can be large up to the fact that choosing an arbitrary solution from this set can, in principle, lead to over-fitting (over general convex problems). Thus, to argue that the algorithm did generalize, further structure in the problem needs to be taken into account. And this is true for *any* regularizer.

Our construction is similar to the previous case in Theorem 3 up to some modification. Therefore, let us show that in the construction above $K_{S,r}$ will be $(T/6, \Theta(1))$-statistically complex. This is less than what we actually desire. We, in fact, observed $T$ examples and not $T/6$. Indeed, in the construction above, we showed that if we project the output of SGD to the observed coordinates, we obtain a solution of the form $(\mathbf{v}_{\pm 1}, \mathbf{v}_{\pm 1}, \cdots, \mathbf{v}_{\pm 1}) \in (\mathbb{R}^2)^T$, where $\mathbf{v}_1, \mathbf{v}_{-1}$ are as in Lemma 1. By projecting this set, it can be seen to be a copy of (up to some rescaling) the normalized unit cube $\mathcal{M} = \{\pm \eta, \pm \eta, \cdots, \pm \eta\} \in \mathbb{R}^T$. This is true since $\|\eta \cdot \mathbf{v}_1 - \eta \cdot \mathbf{v}_{-1}\| = \Theta(\eta)$.

Here, we rely on a construction by Feldman (2016). In order to show that uniform convergence is not equivalent to learnability in the convex optimization setting, Feldman showed (in our terminology) that the set $\mathcal{M} \in \mathbb{R}^T$ is $(T/6, 1/4)$ statistically complex, if $\eta = \Theta(1/\sqrt{T})$.

As discussed, this is less than what we want, as we actually want a set that is at least $(T, \Theta(1))$ statistically complex. To tackle this, on each iteration we show the learner a loss function over multiple pairs of coordinates. Namely, if in the example above we drew at each iteration $f(\mathbf{w}; z)$ where $z \sim D$, now in each iteration we show the algorithm $\frac{1}{k} \sum_{i=1}^{k} f(\mathbf{w}; z_i)$, where $z_i$ are i.i.d. This will reduce the step-size on each coordinate a little bit but if $k$ is constant we will still present a constant loss. On the other hand, now projecting on observed coordinates, SGD will converge to a solution in $(\mathbf{v}_{\pm 1}, \mathbf{v}_{\pm 1}, \ldots, \mathbf{v}_{\pm 1}) \in (\mathbb{R}^2)^{\Theta(kT)}$. Thus we only need a constant $k > 6$ so that the algorithm will converge to a $(T, \Theta(1))$-statistically complex set.

## 5.3 Implicit Bias in Constant Dimension

We next provide a construction in $\mathbb{R}^2$ that again rules out a class of regularizers, in particular strongly convex regularizers (and more generally, strictly quasi-convex regularizers).

In a similar fashion to previous constructions, we make SGD choose from a set of solutions, that exhibit comparable empirical loss. While the dimension of previous constructions depended on $T$, this construction does not. However, for the construction we relax the assumption that $f(\mathbf{w}; z)$ are convex, but $F$ remains convex. Note that the learning guarantees of SGD are completely applicable to this setting.

Our construction relies on a 2-dimensional square, centered at the origin. Inside the square, SGD makes a simple 2-dimensional random walk, while when it exits from the square, it continues to perform a random walk in just one dimension (denoted as $y$), while the other coordinate (denoted as $x$) remains the same. As a result, the optimizer of $F_S$ is independent of $w_x$.

We study the event that $\mathbf{w}$ will stay inside the square for enough iterations to ensure that the variance of $w_x$ will be larger than some constant, but eventually $\mathbf{w}$ exit from the square to make $F_S$ independent of $w_x$. This will result with a set of solutions that share the same empirical error and also SGD can converge to each one of them.

## Footnotes

[1]Since our main focus in this paper is on impossibility results, fixing the Lipschitz constant and the diameter both to 1 does not harm the generality of the setup.

[2]In fact, the proofs will be significantly simpler; for example, in the proof overview we actually consider the last iterate for simplicity.

[3]We note though that it can be turned to an affine invariant optimization algorithm (Koren and Livni, 2017).

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

# A   Technical Preliminaries

## A.1   Distance Functions

Throughout, we will repeatedly use functions of the following form in our constructions:

$$f_{A,\Sigma}(\mathbf{w}) = \frac{1}{2} \min_{v \in A} \left\{ (\mathbf{w} - \mathbf{v})^\top \Sigma (\mathbf{w} - \mathbf{v}) \right\}, \tag{9}$$

where $A$ is a set of the form $A = \{(\alpha, \theta) : -b_1 < \alpha < b_2\}$, $\theta \in \mathbb{R}$, $b_1, b_2 \in \mathbb{R}_+$ and $\Sigma$ is a PSD matrix of the following form

$$\Sigma = \begin{bmatrix} \sigma^2 & \sigma/2 \\ \sigma/2 & 1 \end{bmatrix}.$$

Because $A$ is convex, it is known that a function $f$ of the aforementioned form, depicted in Eq. (9), is indeed a convex function (see Boyd and Vandenberghe (2004) example 3.16). If there is no reason for confusion we will omit dependence in $A$ and $\Sigma$ and simply write $f(\mathbf{w})$.

It can be seen that for a function $f_{A,\Sigma}$ of the form in Eq. (9), the gradient is given by

$$\nabla f(\mathbf{w}) = \Sigma(\mathbf{w} - \mathbf{v}(\mathbf{w})),$$

where we denote $\mathbf{v}(\mathbf{w}) = \arg\min_{v \in A} (\mathbf{w} - \mathbf{v})^\top \Sigma (\mathbf{w} - \mathbf{v})$. As a corollary one can obtain the following expressions for the gradient

$$\nabla f(\mathbf{w}) = \begin{cases} \Sigma(\mathbf{w} - (b_1, \theta)) & w_1 + \frac{1}{2\sigma}(w_2 - \theta) < b_1 \\ \begin{pmatrix} 0 \\ \frac{3(w_2 - \theta)}{4} \end{pmatrix} & b_1 < w_1 + \frac{1}{2\sigma}(w_2 - \theta) < b_2 \\ \Sigma(\mathbf{w} - (b_2, \theta)) & b_2 \leq w_1 + \frac{1}{2\sigma}(w_2 - \theta) \end{cases} \tag{10}$$

## A.2   Feldman's Statistically Complex Set

A key technical tool in the proof of Theorem 4 is a construction by Feldman, Feldman (2016), of a statistically complex set in $\mathbb{R}^d$. While Feldman's construction is not the first to show that the sample complexity of an ERM algorithm may scale with the dimension, it greatly improved over previous construction Shalev-Shwartz et al. (2009), and showed that the dependence may be *linear* in the dimension.

We will exploit here Feldman's set in order to construct an example where SGD essentially picks arbitrarily an element from a statistically complex set, akin to ERM, and we will need the following statement due to Feldman

**Theorem 6** (Essentially Theorem 3.3 in Feldman, 2016). *Let* $\mathcal{W}_d = \{-\frac{1}{\sqrt{d}}, \frac{1}{\sqrt{d}}\}^d$. *There exists a distribution $D$ over $1$-Lipschitz convex functions such that given a sample $|S| < d/6$ drawn i.i.d from $D$ then w.p. $1/2$ (over the sample $S$) there exists $\mathbf{w} \in \mathcal{W}_d$ such that*

$$\frac{1}{|S|} \sum_{t=1}^{|S|} f(\mathbf{w}, z_t) = 0, \tag{11}$$

*but*

$$\mathbb{E}_{z \sim D}[f(\mathbf{w}, z)] = 1/4. \tag{12}$$

We will need a slightly stronger version of the theorem which is an immediate corollary

**Corollary 6.1.** *Let $A \subseteq \mathcal{W}_d$, such that $|A| \geq 2^{d-1}$, then $A$ is $(d/6, 1/4)$-statistically complex.*

*Proof.* For two vectors $\mathbf{v} \in \{-1, 1\}^d$ and an element $\mathbf{w} \in \mathcal{W}_d$ let $\mathbf{v} * \mathbf{w} \in \mathcal{W}_d$ be the pointwise product between $\mathbf{w}$ and $\mathbf{v}$, i.e.

$$(\mathbf{v} * \mathbf{w})_i = \mathbf{v}_i \cdot \mathbf{w}_i.$$

Let $D$ be the distribution from Theorem 6 and consider a distribution where we draw uniformly an elements $\mathbf{v} \in \{-1, 1\}^d$ and a sample $S$ of size d/6 i.i.d from $D$. One can show that with probability $|A|/(2^{d+1})$ we have that there exists an elements $\mathbf{w} \in A$ such that

$$\frac{1}{|S|} \sum_{t=1}^{|S|} f(\mathbf{v} * \mathbf{w}; z_t) = 0 \tag{13}$$

but

$$\mathbb{E}_{z \sim D} f(\mathbf{v} * \mathbf{w}; z) = 1/4. \tag{14}$$

In particular, there exists a $\mathbf{v}$ such that with probability $\frac{|A|}{2^{d+1}}$, Eqs. (13) and (14) holds for some $\mathbf{w} \in A$ over the random sample $S$. Thus, we can define a convex Lipschitz mapping parameterized by $\mathbf{z}$ such that

$$f_{\mathbf{v}}(\mathbf{w}; z) = f(\mathbf{v} * \mathbf{w}; z).$$

From the above discussion if we draw $z \sim D$ we can see that this distribution demonstrates that $A$ is (d/6,1/4)-statistically complex

$\square$

## A.3 Berry-Esseen Theorem

A very important and valuable tool for analysing the behavior of random walks that we will use is the well-known Berry-Esseen Theorem, discovered independently in Berry (1941); Esseen (1942).

**Theorem 7** (Berry Esseen Theorem). *Let $X_1, X_2, \ldots, X_T$ be zero mean and independent random variables, with $\mathbb{E}(X_i^2) = \sigma_i^2$ and $\mathbb{E}(|X_i^3|) = \rho_i$. Let $S_T = \frac{1}{\sqrt{\sum_{i=1}^{T} \sigma_i^2}} \sum_{i=1}^{T} X_i$, then we have*

$$|P(S_T \leq a) - \Phi(a)| \leq C_{BE} (\sum_{i=1}^{T} \sigma_i^2)^{-3/2} \sum_{i=1}^{T} \rho_i,$$

*where $C_{BE} < 1$ is an absolute constant, and $\Phi(a)$ is the CDF of a unit variate zero-mean Gaussian random variable.*

For a bound $C_{\mathrm{BE}} < 1$ of the absolute constant see, for example, van Beek (1972). We will need the following technical Lemma which is derived via Theorem 7:

**Lemma 2.** *Let $k \geq 0$, and assume $T > 2 \cdot k$. If $X_t$ is a random variables such that*

$$X_t = \begin{cases} c\frac{T-t}{T} & w.p. \ 1/4 \\ -c\frac{T-t}{T} & w.p. \ 1/4 \ , \\ 0 & w.p. \ 1/2 \end{cases}$$

*and $I = \{1, 2, \ldots, T/k\}$, then*

$$P\left(\left|\frac{1}{\sqrt{T}} \sum_{i \in I} X_i\right| < a\frac{c}{\sqrt{50k}}\right) \leq erf(a) + \sqrt{\frac{50^3 k}{T}},$$

*where $erf(a) = \Phi(a) - \Phi(-a)$ is the error function.*

*Proof.* First, we lower bound $\sum_{i \in I} \sigma_i^2$, and obtain that:

$$\sum_{i \in I} E[|X_i|^2] = \frac{c^2}{2} \sum_{i \in I} \left(\frac{T-t}{T}\right)^2$$

$$\geq \frac{c^2}{2T^2} \sum_{t=1}^{T/k} (T-t)^2$$

$$= \frac{c^2}{2T^2} \sum_{t=0}^{T/k-1} \left(\left(\frac{k-1}{k}\right)T + t\right)^2$$

$$\geq \frac{c^2}{2T^2} \max\left\{\frac{(k-1)^2}{k^2}T^2\frac{T}{k}, \sum_{t=0}^{T/k-1} t^2\right\}$$

We also have that for $T > 2 \cdot k$:

$$\sum_{t=0}^{T/k-1} t^2 = \frac{T/k \, (T/k - 1) \, (2T/k - 1)}{6} \geq \frac{T^3}{12k^3}$$

Taken together we obtain that

$$\sum_{i \in I} \mathbb{E}[|X_i|^2] \geq \frac{c^2 T}{2} \max\left\{ \frac{(k-1)^2}{k^2}, \frac{1}{12k^2} \right\}$$

$$\geq \frac{c^2 T}{50k}$$

Next, we lower bound $\sum \rho_i$:

$$\sum_{i \in I} \mathbb{E}[|X_i|^3] \leq \frac{1}{2} \sum_{t=1}^{T/k} \mathbb{E}[|c \frac{(T-t)}{T}|^3] = \frac{c^3}{2T^3} \sum_{t=1}^{T/k} (T-t)^3 \leq \frac{c^3 T}{2T^3 k} T^3 \leq \frac{c^3 T}{2k}$$

Taken together we obtain that

$$P\left( \left| \frac{1}{\sqrt{T}} \sum X_i \right| < a \frac{c}{\sqrt{50k}} \right) \leq P\left( \left| \frac{1}{\sqrt{\sum_{i=1}^{T} \sigma_i^2}} \sum X_i \right| < a \right)$$

$$\leq \Phi(a) - \Phi(-a) + 2 \left( \sum_{i=1}^{T} \sigma_i^2 \right)^{-3/2} \sum_{i=1}^{T} \rho_i$$

$$\leq \Phi(a) - \Phi(-a) + \frac{(50k)^{3/2} c^3 T}{c^3 T^{3/2} k}$$

$$\leq \Phi(a) - \Phi(-a) + \sqrt{\frac{50^3 k}{T}}$$

$\square$

# B    Proofs: Distribution Independent Regularizers

As discussed, the main technical difficulty behind the distribution-independent-regularization results is the existence of a problem instance where GD does not converge to the minimal norm solution:

**Theorem 8** (GD doesn't converge to closest Euclidean norm solution)**.** *For every $0 < \theta_2 \leq 1$, and $0 < \theta_1 \leq 0.025 \cdot \theta_2$, there exists a positive, convex and smooth function $f_{\theta_1, \theta_2}$ such that for $0 < \eta < \frac{1}{3}$, GD outputs $\mathbf{w}_S$ such that*

$$\|\mathbf{w}_S - (\theta_1, \theta_2)\| < \frac{120}{\eta T}$$

*but*

$$f_{\theta_1, \theta_2}((0, \theta_2)) = f_{\theta_1, \theta_2}(\theta_1, \theta_2) = 0$$

*Proof.* For $0 \leq \theta_2 \leq 1$ and $0 \leq \theta_1 \leq 0.025 \cdot \theta_2$ let us define the set: $A_{\theta_1, \theta_2} = \{(\alpha, \theta_2) : 0 \leq \alpha \leq \theta_1\}$. In turn, we define the function $f_{\theta_1, \theta_2}(\mathbf{w})$ to be:

$$f_{\theta_1, \theta_2}(\mathbf{w}) = \arg \min_{v \in A_{\theta_1, \theta_2}} \frac{1}{2} (\mathbf{w} - \mathbf{v})^T \Sigma (\mathbf{w} - \mathbf{v}), \qquad (15)$$

where we let,

$$\Sigma = \begin{pmatrix} 1 & 0.5 \\ 0.5 & 1 \end{pmatrix}.$$

We begin with a proposition that describes the trajectory of GD over the function $f_{\theta_1, \theta_2}$.

**Lemma 3.** *Let* $\mathbf{w}^{(1)}, ..., \mathbf{w}^{(T)}$ *be the sequence defined by running GD over* $f_{\theta_1,\theta_2}$ *with step size* $\eta \leq \frac{1}{3}$, *and $T$ iteration. Then there exist* $\frac{1}{2\cdot\eta} \leq t_0 \leq \frac{3}{\eta}, t_0 \leq t_1 \leq t_0 + \frac{7}{\eta}$ *s.t.*

$$\text{For } 1 \leq t \leq t_0: \quad \boldsymbol{w}^{(t)} = \left(I - (I - \eta\Sigma)^{t-1}\right)\xi_0 \qquad\qquad \text{where } \xi_0 = (0, \theta_2) \qquad (16)$$

*next,*

$$\text{For } t_0 < t \leq t_1: \quad \mathbf{w}^{(t)} = \begin{pmatrix} w_1^{(t_0)} \\ \left(1 - \frac{3\eta}{4}\right)^{t-t_0}\left[w_2^{(t_0)} - \theta_2\right] + \theta_2 \end{pmatrix} \qquad\qquad (17)$$

*and,*

$$\text{For } t_1 < t \leq T: \quad \boldsymbol{w}^{(t)} = \left(I - (I - \eta\Sigma)^{t-t_1}\right)\cdot\xi_1 + (I - \eta\Sigma)^{t-t_1}\cdot\mathbf{w}^{(t_1)} \qquad \text{where } \xi_1 = (\theta_1, \theta_2) \qquad (18)$$

This trajectory is depicted in Fig. 1 for different values of $\theta_1$. We now show how to derive Theorem 8 from Lemma 3 and deter the proof of Lemma 3 to the end of the section. Note that, by notation above, to prove Theorem 8, we only need to show that $\|w_S - \xi_1\| \leq \frac{120}{\eta T}$— the fact that $f_{\theta_1,\theta_2}(0, \theta_2) = f_{\theta_1,\theta_2}(\theta_1, \theta_2)$ is immediate from the definition of $f$.

We will first need to bound the sizes $\|\xi_0\|, \|\xi_1\|, \|\mathbf{w}^{(t)}\|$. One can easily observe that $\|\xi_1\|, \|\xi_0\| < 1.5$. Following the trajectory path of $\mathbf{w}^{(t)}$, provided in Lemma 3 we can also provide a bound on $\mathbf{w}^{(t)}$:

Namely,

- If $t \leq t_0$ we have that $\|\mathbf{w}^{(t)}\| < \|\xi_0\| \leq 1$

- If $t_0 \leq t \leq t_1$, then $\|\mathbf{w}^{(t)}\| \leq \|\mathbf{w}^{(t_0)}\| + \theta_2 \leq 2$.

- And if $t \geq t_1$ we have that $\|\mathbf{w}^{(t)}\| \leq \|\mathbf{w}^{(t_1)}\| + \|\xi_1\| < 5$

Taken together we have that $\|\mathbf{w}^{(t)}\| < 5$.

Finally, by simple calculation we can show that the singular values of $\Sigma$ are $3/2$ and $1/2$. Hence,

$$\| (I - \eta\Sigma) \|_\infty \leq (1 - \frac{\eta}{2}). \qquad\qquad (19)$$

where $\| \cdot \|_\infty$ denotes the spectral norm of a matrix. We are now ready to show that $\mathbf{w}_S$ converges to $\xi_1$:

$$\|\mathbf{w}_S - \xi_1\|_2 \leq \frac{1}{T}\sum_{t=1}^{T}\|(\mathbf{w}^{(t)} - \xi_1)\|_2 = \frac{1}{T}\sum_{t=1}^{t_1}\|(\mathbf{w}^{(t)} - \xi_1)\|_2 + \frac{1}{T}\sum_{t=t_1+1}^{T}\|(\mathbf{w}^{(t)} - \xi_1)\|_2$$

$$\leq \frac{10t_1}{T} + \frac{1}{T}\sum_{t=t_1+1}^{T}\|\mathbf{w}^{(t)} - \xi_1\|_2 \qquad\qquad \|\mathbf{w}^{(t)}\|, \|\xi_1\| < 5$$

$$= \frac{100}{\eta T} + \frac{1}{T}\sum_{t=t_1+1}^{T}\|(1 - \eta\Sigma)^{t-t_1}\cdot(\mathbf{w}^{(t_1)} - \xi_1)\| \qquad t_1 < 10/\eta, \ Eq. (18)$$

$$\leq \frac{100}{\eta T} + \frac{1}{T}\sum_{t=t_1+1}^{T}\|(I - \eta\Sigma)^{t-t_1}\|_\infty\cdot\|\mathbf{w}^{(t_1)} - \xi_1\|_2$$

$$\leq \frac{100}{\eta T} + \frac{10}{T}\sum_{t=1}^{T-t_1}\|(I - \eta\Sigma)\|_\infty^t \qquad\qquad \|\mathbf{w}^{(t_1)} - \xi_1\| < 10$$

$$\leq \frac{100}{\eta T} + \frac{10}{T}\sum_{t=0}^{\infty}(1 - \frac{\eta}{2})^t \qquad\qquad Eq. (19)$$

$$\leq \frac{100}{\eta T} + \frac{10}{T}\cdot\frac{2}{\eta} \leq \frac{120}{\eta T}$$

$$\square$$

**Proof of Lemma 3** First note that $f_{\theta_1,\theta_2}$ is a function of the form depicted in Eq. (9), with parameter $\sigma = 1$. We thus obtain two boundary conditions that governs the behavior of the trajectory:

$$w_1 + \frac{1}{2}w_2 < \frac{1}{2}\cdot\theta_2 \qquad\qquad (20)$$

$$w_1 + \frac{1}{2}w_2 \geq \frac{1}{2} \cdot \theta_2 + \theta_1, \tag{21}$$

Given $\eta$, we claim that Lemma 3 holds if we let $t_0$ denote the first iterate such that $\mathbf{w}^{(t_0)}$ violates Eq. (20), when running GD, and if $t_1$ denotes the first iterate for which $\mathbf{w}^{(t)}$ satisfies Eq. (21). We will split the proof into 3 parts, according to GD's trajectory, i.e. $t \leq t_0, t_0 < t \leq t_1, t > t_1$

**Claim 9.** *There exists $\frac{1}{2 \cdot \eta} \leq t_0 \leq \frac{3}{\eta}$ such that $\mathbf{w}^{(t_0)}$ is the first iterate that violates Eq. (20). Further, for any $t \leq t_0$, $\mathbf{w}^{(t)}$ can be calculated by Eq. (16). And finally, $0.03\theta_2 \leq w_1^{(t_0)}$.*

*Proof.* First note that $\mathbf{w}^{(1)}$ satisfies Eq. (20), hence $t_0 \geq 1$. Now, following the calculation of the derivative provided in, Eq. (10) we obtain the following update step: $\mathbf{w}^{(t+1)} = \mathbf{w}^{(t)} - \eta\Sigma\left(\mathbf{w}^{(t)} - \xi_0\right)$. which we can rewrite as:

$$\mathbf{w}^{(t+1)} = (I - \eta\Sigma)\mathbf{w}^{(t)} + \eta\Sigma\xi_0. \tag{22}$$

By induction one can show that for $2 \leq t \leq t_0$:

$$\mathbf{w}^{(t)} = \sum_{i=0}^{t-2}(I - \eta\Sigma)^i \cdot (\eta\Sigma\xi_0)$$

$$= \left(I - (I - \eta\Sigma)^{t-1}\right)\xi_0. \tag{23}$$

This shows that for any $t \leq t_0$, $\mathbf{w}^{(t)}$ can be calculated by Eq. (16). We proceed with the proof to show that $\frac{1}{2\eta} \leq t_0 \leq \frac{3}{\eta}$,

Considering the singular value decomposition of $\Sigma$ one can show that:

$$(I - \eta\Sigma)^{t-1} = \frac{1}{2}\begin{pmatrix} [(1 - 3\eta/2)^{t-1} + (1 - \eta/2)^{t-1}] & [(1 - 3\eta/2)^{t-1} - (1 - \eta/2)^{t-1}] \\ [(1 - 3\eta/2)^{t-1} - (1 - \eta/2)^{t-1}] & [(1 - 3\eta/2)^{t-1} + (1 - \eta/2)^{t-1}] \end{pmatrix}. \tag{24}$$

Plugging this in Eq. (23), we obtain that for any $t \leq t_0$:

$$\mathbf{w}^{(t)} = \frac{\theta_2}{2}\begin{pmatrix} [(1 - \eta/2)^{t-1} - (1 - 3\eta/2)^{t-1}] \\ 2- & [(1 - \eta/2)^{t-1} + (1 - 3\eta/2)^{t-1}] \end{pmatrix} \tag{25}$$

To obtain the lower bound on $t_0$ observe that $t_0$ satisfies:

$$w_1^{(t_0)} + \frac{1}{2}(w_2^{(t_0)} - \theta_2) \geq 0,$$

Plugging Eq. (25) and dividing by $\theta_2/2$ we obtain that:

$$\left(1 - \frac{\eta}{2}\right)^{t_0-1} - \left(1 - \frac{3\eta}{2}\right)^{t_0-1} + \frac{1}{2}\left(2 - \left(1 - \frac{\eta}{2}\right)^{t_0-1} - \left(1 - \frac{3\eta}{2}\right)^{t_0-1} - 2\right) \geq 0.$$

Rearranging terms we get:

$$\frac{1}{2}\left(1 - \frac{\eta}{2}\right)^{t_0-1} - \frac{3}{2}\left(1 - \frac{3\eta}{2}\right)^{t_0-1} \geq 0$$

Which for $\eta < 1/3$, can be rewritten as:

$$\left(1 + \frac{2\eta}{2 - 3\eta}\right)^{t_0-1} = \left(\frac{2 - \eta}{2 - 3\eta}\right)^{t_0-1} \geq 3. \tag{26}$$

This leads to

$$
\begin{aligned}
t_0 &\geq \frac{1}{\ln(1 + \frac{2\eta}{2-3\eta})} && \ln(3) \geq 1 \\
&\geq \frac{2 - 3\eta}{2\eta} && \ln(x + 1) \leq x \\
&= \frac{1}{\eta} - \frac{3}{2} \\
&\geq \frac{1}{2\eta} && \eta \leq \frac{1}{3}
\end{aligned}
$$

Next we provide an upper bound for $t_0$. Again, for every $t < t_0$ Eq. (20) is satisfied, which, as we already saw ( in Eq. (26)) means that for every $t < t_0$:

$$\forall t < t_0, \quad \left(1 + \frac{2\eta}{2 - 3\eta}\right)^{t-1} \leq 3. \tag{27}$$

Using the inequality $(1 + 2/n)^n \geq 3$, we obtain

$$\left(1 + \frac{2\eta}{2 - 3\eta}\right)^{t-1} \geq (1 + \eta)^{t-1} \geq 3^{\frac{\eta}{2}(t-1)}$$

In particular for $t \geq \frac{2}{\eta} + 1$ Eq. (27) is violated and hence $t_0 \leq \frac{3}{\eta}$.

Finally, we provide a lower bound for $w_1^{(t_0)}$. Namely, we want to show that $w_1^{(t_0)} \geq 0.04\theta_2$.

First, by rearranging terms at Eq. (26) we obtain that $t_0$ is sufficiently large so that $\left(1 - \frac{\eta}{2}\right)^{t_0-1} \geq 3\left(1 - \frac{3\eta}{2}\right)^{t_0-1}$. Again applying the formula for $\mathbf{w}^{(t_0)}$ in Eq. (25) we have that:

$$
\begin{aligned}
w_1^{(t_0)} = \frac{\theta_2}{2} \cdot [(1 - \frac{1}{2} \cdot \eta)^{t_0-1} - (1 - \frac{3}{2} \cdot \eta)^{t_0-1}] &\geq \frac{\theta_2}{4}\left(1 - \frac{1}{2}\eta\right)^{t_0-1} \\
&\geq \frac{\theta_2}{4}\left(1 - \frac{1}{2}\eta\right)^{3/\eta} & t_0 < \frac{3}{\eta} \\
&\geq 2^{-5}\theta_2 & \left(1 - \frac{1}{2n}\right)^n > \frac{1}{2} \quad (28)
\end{aligned}
$$

This concludes the analysis of the first phase of the trajectory. We next move on to the case $t_0 \leq t \leq t_1$ □

**Claim 10.** *Let $t_0 \leq t \leq t_1$. Then $\boldsymbol{w}^{(t)}$ can be calculated by Eq. (17). Moreover $t_1 \leq t_0 + \frac{7}{\eta}$.*

*Proof.* We again apply the calculation of the derivative provided in Eq. (10) at $t_0 \leq t \leq t_1$ and obtain :

$$\nabla f(\mathbf{w}^{(t)}) = \begin{pmatrix} 0 \\ \frac{3}{4}(w_2^{(t)} - \theta_2) \end{pmatrix}. \tag{29}$$

Note that this proves that $w_1^{(t)} = w_1^{(t_0)}$. For $w_2^{(t)}$, we have that

$$w_2^{(t)} = w_2^{(t-1)}(1 - \frac{3}{4}\eta) + \frac{3}{4} \cdot \eta \cdot \theta_2,$$

which leads by induction to the following:

$$
\begin{aligned}
w_2^{(t)} &= \left(1 - \frac{3}{4}\eta\right)^{t-t_0} w_2^{(t_0)} + \sum_{i=0}^{(t-t_0)-1} \left(1 - \frac{3}{4}\eta\right)^i \cdot \frac{3\eta\theta_2}{4} \\
&= \left(1 - \frac{3}{4}\eta\right)^{t-t_0} w_2^{(t_0)} + \left(1 - \left(1 - \frac{3}{4}\eta\right)^{t-t_0}\right)\theta_2 \\
&= \left(1 - \frac{3}{4}\eta\right)^{t-t_0} \left[w_2^{(t_0)} - \theta_2\right] + \theta_2.
\end{aligned}
$$

This shows that for any $t_0 \leq t \leq t_1$ Eq. (17) holds.

We next bound $t_1$. Recall that $t_1$ is defined to be the first iterate for which Eq. (21) is satisfied. Let us show that for any $t$ s.t $t_0 + \frac{7}{\eta} < t$ holds, Eq. (21) is satisfied and hence $t_1 \leq t_0 + 7/\eta$. Equivalently we will show that for $t > t_0 + 7/\eta$, the following equation holds:

$$\theta_2 - w_2^{(t)} \leq 2(w_1^{(t)} - \theta_1). \tag{30}$$

Indeed, let $t < t_1$, then

$$2 \cdot (w_1^{(t)} - \theta_1) = 2 \cdot (w_1^{(t_0)} - \theta_1) \qquad\qquad (w_1^{(t)} = w_1^{(t_0)} \text{ by } Eq.\ (17))$$
$$\geq 2 \cdot (2^{-5} \cdot \theta_2 - \theta_1) \qquad\qquad (w_1^{(t_0)} \geq 2^{-5}\theta_2 \text{ by } Eq.\ (28))$$
$$\geq 2 \cdot (2^{-5} \cdot \theta_2 - 0.025\theta_2) \qquad\qquad (\theta_1 \leq 0.025 \cdot \theta_2)$$
$$\geq 0.01 \cdot \theta_2$$

Next assume that $t \geq t_0 + \frac{7}{\eta}$, then

$$0.01 \cdot \theta_2 \geq e^{-\frac{3\eta \cdot (t-t_0)}{4}} \theta_2 \qquad\qquad t \geq t_0 + \frac{20}{3\eta}$$
$$\geq \left(1 - \frac{3}{4}\eta\right)^{t-t_0} \theta_2$$
$$\geq \left(1 - \frac{3}{4}\eta\right)^{t-t_0} \left[\theta_2 - w_2^{(t_0)}\right] \qquad\qquad (w_2^{(t_0)} \geq 0)$$
$$= \theta_2 - \mathbf{w}_2^{(t)}. \qquad\qquad Eq.\ (17)$$

We now move to the last phase of the trajectory. $\qquad\qquad\qquad\qquad\qquad\qquad\qquad \square$

**Claim 11.** *Let $t \geq t_1$, then $\boldsymbol{w}^{(t)}$ can be calculated by Eq. (18).*

*Proof.* Let $t \geq t_1$ be such that Eq. (21) holds. Then again, we consider the formula of the derivative $\nabla f(\mathbf{w})$ (see Eq. (10)) and have that
$$\nabla f(\mathbf{w}) = \Sigma(\mathbf{w} - \xi_1).$$

We obtain the following recursive formula for $t$ if Eq. (21) holds for all $t_1 \leq t' \leq t$:

$$\mathbf{w}^{(t)} = (I - \eta\Sigma)\mathbf{w}^{(t-1)} + \eta\Sigma\xi_1$$
$$= (I - \eta\Sigma)^{t-t_1}\mathbf{w}^{(t_1)} + \sum_{i=0}^{t-t_1-1}(I - \eta\Sigma)^i \eta\Sigma\xi_1$$
$$= (I - \eta\Sigma)^{t-t_1}(\mathbf{w}^{(t_1)} - \xi_1) + \xi_1 \qquad\qquad (31)$$

This shows that $\mathbf{w}^{(t)}$ can be calculated via Eq. (18). It remains thus to show that for any $t \geq t_1$, Eq. (21) always holds. We prove this by induction. Note that for the base case, this follows from the definition of $t_1$. We can thus assume by induction hypothesis that $\mathbf{w}^{(t)}$ satisfies Eq. (31), and we want to prove that

$$w_1^{(t)} + 1/2w_2^{(t)} - \theta_1 - 1/2\theta_2 \geq 0.$$

For succinctness, let us write

$$\alpha_t = \left(1 - \frac{3\eta}{2}\right)^{t-t_1}, \quad \text{and,} \quad \beta_t = \left(1 - \frac{\eta}{2}\right)^{t-t_1}.$$

We will denote also $\mathbf{v} = \begin{pmatrix} 1 \\ 1/2 \end{pmatrix}$ Then using Eq. (24) and Eq. (31) we have that

$$w_1^{(t)} + 1/2w_2^{(t)} - \theta_1 - 1/2\theta_2 = \mathbf{v}^\top(\mathbf{w}^{(t)} - \xi_1)$$
$$= \mathbf{v}^\top(1 - \eta\Sigma)^{t-t_1}(\mathbf{w}^{(t_1)} - \xi_1) \qquad\qquad Eq.\ (31)$$
$$= (3/2\alpha_t + 1/2\beta_t)(\mathbf{w}_1^{(t_1)} - \theta_1) + (3/2\alpha_t - 1/2\beta_t)(\mathbf{w}_2^{(t_1)} - \theta_2) \qquad\qquad Eq.\ (24)$$
$$\geq (3/4\alpha_t - 3/4\beta_t)(\mathbf{w}_2^{(t_1)} - \theta_2)) \qquad\qquad 2(\mathbf{w}_1^{(t_1)} - \theta_1) \geq \theta_2 - \mathbf{w}_2^{(t_1)}$$
$$\geq 0$$

where the last inequality is true since $\alpha_t \leq \beta_t$ for $\eta < 1/3$ and we also have that $\mathbf{w}_2^{(t_1)} < \theta_2$. $\qquad \square$

This concludes the proof of Lemma 3.

## B.1  Proof of Theorem 1

Theorem 1 is an almost immediate corollary of Theorem 8. Indeed, let $r$ be a strongly convex function. Take $\mathbf{e}_2 = (0,1)$ and let $\mathbf{w}_0 = (0.024, 1)$. Consider the set $a_{(\mathbf{e}_2, \mathbf{w}_0)} = \{\alpha \mathbf{e}_2 + (1 - \alpha)\mathbf{w}_0 : 0 \le \alpha \le 1\}$. and let

$$\mathbf{w}^* = \arg\min_{\mathbf{w} \in a_{\mathbf{e}_2, \mathbf{w}_0}} r(\mathbf{w}).$$

Now we either have $\|\mathbf{w}^* - \mathbf{e}_2\| \ge 0.012$, or $\|\mathbf{w}^* - \mathbf{w}_0\| \ge 0.012$. We will show that in both cases we can choose a function $f \ge 0$, such that $f(\mathbf{e}_2) = f(\mathbf{w}_0) = f(\mathbf{w}^*) = 0$ and that $\mathbf{w}_S$ will satisfy

$$\|\mathbf{w}^* - \mathbf{w}_S\| > 0.01 \tag{32}$$

and that if $\Pi(\mathbf{w}_S)$ is the projection of $\mathbf{w}_S$ on $a_{\mathbf{e}_2, \mathbf{w}_0}$ then

$$\|\mathbf{w}_S - \Pi(\mathbf{w}_S)\| < \frac{120}{T\eta} \tag{33}$$

This will conclude the proof. Indeed, by strong convexity:

$$
\begin{aligned}
r(\mathbf{w}_S) &\ge r(\mathbf{w}^*) + \nabla r(\mathbf{w}^*)^\top (\mathbf{w}_S - \mathbf{w}^*) + \lambda \|\mathbf{w}^* - \mathbf{w}_S\|^2 \\
&\ge r(\mathbf{w}^*) + \nabla r(\mathbf{w}^*)^\top (\Pi(\mathbf{w}_S) - \mathbf{w}^*) + \nabla r(\mathbf{w}^*)^\top (\mathbf{w}_S - \Pi(\mathbf{w}_S)) + \lambda \|\mathbf{w}^* - \mathbf{w}_S\|^2 \\
&\ge r(\mathbf{w}^*) - \frac{120}{T\eta} + \lambda \|\mathbf{w}^* - \mathbf{w}_S\|^2 \qquad\qquad\qquad \text{Lipschitness of } r \\
&\ge r(\mathbf{w}^*) - \frac{120}{T\eta} + 10^{-6}\lambda
\end{aligned}
$$

We are thus left with proving the existence of $f$ in each case:

- First assume that $\|\mathbf{w}^* - \mathbf{e}_2\| > 0.012$. We will construct a distribution s.t. $\mathbf{w}^{(t)}$ will converge to $\mathbf{e}_2$: set

$$f(\mathbf{w}) = \frac{1}{2} \cdot \min_{\mathbf{v} \in a_{(\mathbf{e}_2, \mathbf{w}_0)}} \|\mathbf{w} - \mathbf{v}\|^2$$

  (i.e. $\Sigma = I$). Our distribution $D$ is defined to choose $f$ w.p. 1.

  A simple analysis of the update step of SGD shows that for $\eta < 1$, we have that $\mathbf{w}^{(t+1)} = \sum_{i=0}^{t-1}(1-\eta)^i \eta \mathbf{e}_2$. Hence,

$$
\begin{aligned}
\|\mathbf{w}_S - \mathbf{e}_2\| = \|\frac{1}{T}\sum_{t=1}^{T} \mathbf{w}^{(t)} - \mathbf{e}_{21}\| &\le \frac{1}{T}\sum_{t=1}^{T}\|\mathbf{w}^{(t)} - \mathbf{e}_2\| \\
&= \frac{1}{T}\sum_{t=1}^{T}\|\sum_{i=1}^{t-1}(1-\eta)^i \eta \mathbf{e}_2 - \mathbf{e}_2\| \\
&= \frac{1}{T}\sum_{t=1}^{T}\|(1-\eta)^t \mathbf{e}_2\| \qquad\qquad \sum_{i=1}^{t}(1-\eta)^t = \frac{1 - (1-\eta)^{t+1}}{\eta} \\
&\le \frac{1}{T}\sum_{t=1}^{T}(1-\eta)^t \\
&\le \frac{1}{T\eta}
\end{aligned}
$$

  In particular we have that

$$\|\mathbf{w}_S - \Pi(\mathbf{w}_S)\| < \|\mathbf{w}_S - \mathbf{e}_2\| < \frac{120}{T\eta},$$

  also assuming $\frac{1}{T\eta}$ is sufficiently small we have that $\|\mathbf{w}^* - \mathbf{w}_S\| > 0.01$.

- Next we assume that $\|\mathbf{w}^* - \mathbf{w}_0\| > 0.012$. We now apply Theorem 8. and we let $f = f_{\theta_1, \theta_2}$ be as in Theorem 8 with parameters $\theta_1 = 0.024$ and $\theta_2 = 1$. Then, by Theorem 8 we have that $\|\mathbf{w}_S - \mathbf{w}_0\| < \frac{120}{\eta T}$, and we obtain as before that $\|\mathbf{w}^* - \mathbf{w}_S\| > 0.01$ and that $\|\mathbf{w}_S - \Pi(\mathbf{w}_S)\| < \frac{120}{\eta}$, as required.

## B.2  Proof of Theorem 2

For a vector $\mathbf{w} \in \mathcal{W} \subseteq \mathbb{R}^2$ let us denote by $\mathbf{w}^\perp := (w_2, -w_1)$. In particular, we have that $\mathbf{w}^\top \mathbf{w}^\perp = 0$ and $\|\mathbf{w}\| = \|\mathbf{w}^\perp\|$. Our proof relies on the following claim which we prove at the end of this section.

**Claim 12.** *Let $r$ be an admissible regularizer over $\mathbb{R}^2$. There are two points $\mathbf{w}_1$ and $\mathbf{w}_2$ in the unit ball such that for some $-0.005\|\mathbf{w}_1\| < \delta < 0.005\|\mathbf{w}_1\|$ we have*

$$\mathbf{w}_2 = \mathbf{w}_1 + \delta \mathbf{w}_1^\perp,$$

*and $r(\mathbf{w}_1) \neq r(\mathbf{w}_2)$.*

We next proceed with proof of Theorem 2. Let $\mathbf{w}_1$ and $\mathbf{w}_2$ be as in Claim 12. First, because GD is invariant to rotations, we can assume w.l.o.g that $\mathbf{w}_1 = \|\mathbf{w}_1\| \cdot \mathbf{e}_2$, and hence $\mathbf{w}_2 = (1, \delta)\|\mathbf{w}_1\|$.

We now set $c_r = \frac{1}{2}|r(\mathbf{w}_1) - r(\mathbf{w}_2)|$. To choose $T_r, D_r$ and $\mathbf{w}_r$ we now look at two cases: if $r(\mathbf{w}_1) > r(\mathbf{w}_2)$ and if $r(\mathbf{w}_1) < r(\mathbf{w}_2)$.

- First suppose $r(\mathbf{w}_1) > r(\mathbf{w}_2)$. By upper-semicontinuity there exists a neighborhood $\delta_1$ such that for every $\mathbf{w}$ s.t. $\|\mathbf{w} - \mathbf{w}_1\| < \delta_1$, satisfies $r(\mathbf{w}) > r(\mathbf{w}_2) + c_r$. We thus set $T_r = \frac{1}{\eta\delta_1}$, and $\mathbf{w}_r = \mathbf{w}_2$.

  We are left with choosing $D_r$. Note that in this case, the regularizer prefers a point with large Euclidean norm over a point with smaller Euclidean norm. Thus, to show it is not the implicit bias of SGD we only need to construct a distribution that is biased towards smaller Euclidean norms:

  Indeed, consider the set $a_{(\mathbf{w}_1, \mathbf{w}_2)} = \{\alpha\mathbf{w}_1 + (1-\alpha)\mathbf{w}_2 : 0 \leq \alpha \leq 1\}$ we set

  $$f(\mathbf{w}) = \frac{1}{2} \cdot \min_{\mathbf{v} \in a_{(\mathbf{w}_1, \mathbf{w}_2)}} \|\mathbf{w} - \mathbf{v}\|^2$$

  Our distribution $D_r$ is defined to choose $f$ w.p. 1. Having defined $T_r, c_r, \mathbf{w}_r$ and $D_r$ we now set out to prove the result.

  A simple analysis of the update step of SGD shows that for $\eta < 1$ we have for every $\mathbf{w}^{(t)}$ that $\mathbf{w}^{(t+1)} = \sum_{i=0}^{t-1}(1-\eta)^i \eta \mathbf{w}_1$. Hence,

$$
\begin{aligned}
\|\mathbf{w}_S - \mathbf{w}_1\| = \|\frac{1}{T_r}\sum_{t=1}^{T_r} \mathbf{w}^{(t)} - \mathbf{w}_1\| &\leq \frac{1}{T_r}\sum_{t=1}^{T_r} \|\mathbf{w}^{(t)} - \mathbf{w}_1\| \\
&= \frac{1}{T_r}\sum_{t=1}^{T_r}\|\sum_{i=1}^{t-1}(1-\eta)^i \eta \mathbf{w}_1 - \mathbf{w}_1\| \\
&= \frac{1}{T_r}\sum_{t=1}^{T_r}\|(1-\eta)^t \mathbf{w}_1\| \qquad \sum_{i=1}^{t}(1-\eta)^t = \frac{1-(1-\eta)^{t+1}}{\eta} \\
&\leq \frac{1}{T_r}\sum_{t=1}^{T_r}(1-\eta)^t \\
&\leq \frac{1}{T_r\eta} \\
&= \delta_1 \qquad\qquad\qquad\qquad\qquad\qquad T_r = \frac{1}{\delta_1\eta}
\end{aligned}
$$

  By property of $\delta_1$ we have that $r(\mathbf{w}_S) > r(\mathbf{w}_2) + c_r$.

  But because $\mathbf{w}_2$ is optimal (i.e. attain zero on $f$), we also have $F_S(\mathbf{w}_S) > F(\mathbf{w}_r)$. This proves the case $r(\mathbf{w}_1) > r(\mathbf{w}_2)$.

- Next, assume that $r(\mathbf{w}_1) < r(\mathbf{w}_2)$. As before we have a neighborhood $\delta_2$ such that if $\|\mathbf{w} - \mathbf{w}_2\| < \delta_2$ then we are guaranteed that $r(\mathbf{w}) > r(\mathbf{w}_1) + c_r$. We choose then $T_r = \frac{1}{\delta_2\eta}$ and $\mathbf{w}_r = \mathbf{w}_1$.

  To define $D_r$, we now use the function $f_{\theta_1, \theta_2}$ from Theorem 8. We assume w.l.o.g that $\delta > 0$, if this is not the case we can use that function $f_{\theta_1, \theta_2}(\mathbf{w}) = f_{\theta_1, \theta_2}(-\mathbf{w})$.

  Let us set $\theta_2 = \|\mathbf{w}_1\|$ and $\theta_1 = |\delta|\|\mathbf{w}_1\| < 0.05\theta_2$. Again, we consider a deterministic distribution $D_r$ that chooses $f_{\theta_1, \theta_2}$ w.p. 1.

Recall that we assume that $\mathbf{w}_1 = \|\mathbf{w}_1\|\mathbf{e}_2$, hence $\mathbf{w}_1 = (0, \theta_2)$ and $\mathbf{w}_2 = (\theta_1, \theta_2)$. Hence, by Theorem 8, if we run over a sample of size $T_r > \frac{1}{\delta_2 \eta}$, we obtain that

$$\|\mathbf{w}_S - \mathbf{w}_2\| < \delta_2.$$

In particular $r(\mathbf{w}_S) > r(\mathbf{w}_1) + c_r$. But again $F_S(\mathbf{w}_S) \geq F(\mathbf{w}_1)$, because $\mathbf{w}_1$ is optimal.

**Proof of Claim 12**  First, let us assume that there are $\mathbf{u}, \mathbf{v}$ such that $\|\mathbf{u}\|_2, \|\mathbf{v}\|_2 = a$ and $r(\mathbf{u}) \neq r(\mathbf{v})$ (at the end we will show that for admissible regularizer we always have such two points).

We will also assume that $\|\mathbf{u} - \mathbf{v}\|_2 \leq 10^{-9} \cdot a^2$. If this was not the case we can cover the sphere $\{\mathbf{w} : \|\mathbf{w}\|_2 = a\}$ with balls with radius $10^{-9} \cdot a^2$, and have a constant function at every ball, concluding that $r$ is constant on the sphere (which contradicts our assumption).

Next, we also assume that $|u_2| > \frac{1}{2}a$, (either $|u_1| > \frac{1}{2}a$ or $|u_2| > \frac{1}{2}a$, and the proof is similar in both cases so we will analyse only the later case).
Then, since $u_1^2 + u_2^2 = v_1^2 + v_2^2 = a^2$, one can show that

$$\frac{u_1 - v_1}{v_2 + u_2} = \frac{v_2 - u_2}{u_1 + v_1}.$$

So, by choosing $\delta = \frac{u_1 - v_1}{v_2 + u_2} = \frac{v_2 - u_2}{u_1 + v_1}$ we have that:

$$
\begin{aligned}
|\delta| = \frac{|u_1 - v_1|}{|v_2 + u_2|} = \frac{|u_1 - v_1|}{|v_2| + |u_2|} && (|u_2 - v_2| < 10^{-9}a^2, 0.5a < |u_2|) \\
\leq 4\frac{|u_1 - v_1|}{a} && (|v_2| + |u_2| > a/4) \\
\leq 0.0025a && (|u_1 - v_1| < 0.0025/4 \cdot a)
\end{aligned}
$$

Using the first equality we can show that $u_1 - \delta u_2 = v_1 + \delta v_2$ Similarly we can show that $u_2 + \delta u_1 = v_2 - \delta v_1$. Taken together we obtain that

$$\mathbf{u} + \delta \mathbf{u}^\perp = \mathbf{v} - \delta \mathbf{v}^\perp.$$

In particular $r(\mathbf{u} + \delta \mathbf{u}^\perp) = r(\mathbf{v} - \delta \mathbf{v}^\perp)$. Since $r(\mathbf{u}) \neq r(\mathbf{v})$, we either have $r(\mathbf{u}) \neq r(\mathbf{u} + \delta \mathbf{u}^\perp)$, or $r(\mathbf{v}) \neq r(\mathbf{v} - \delta \mathbf{v}^\perp)$. In the former case we choose $\mathbf{w}_1 = \mathbf{u}$, whereas in the latter case we choose $\mathbf{w}_1 = \mathbf{v}$.

Finally, so far we assume we can find two points on a sphere with different regularization penalty. Next, we assume that on every sphere $r$ is constant. Assume also to the contrary that for every $-0.0025\|\mathbf{w}\| < \delta < 0.0025\|\mathbf{w}\|$:

$$r(\mathbf{w} + \delta \mathbf{w}^\perp) = r(\mathbf{w}).$$

It is not hard to show that in this case $r$ is constant everywhere except maybe 0, making it in-admissible.

# C  Proofs II: Distribution Dependent Regularization

We start this section by proving the existence of the auxiliary construction in Lemma 1

**Lemma 1.** *For every constant $0 < c < 1$, there are two 1-Lipschitz functions $f(\mathbf{w}; \pm 1)$ over $\mathbb{R}^2$ such that if $\mathbf{v}_1 = -\nabla f(0; 1)$ and $\mathbf{v}_{-1} = -\nabla f(0; -1)$ and $c < \frac{1}{2}\eta < 1$ then $\|\mathbf{v}_1 - \mathbf{v}_{-1}\| \geq 1/4$ and $f(\eta \mathbf{v}_1; z) = f(\eta \mathbf{v}_{-1}; z)$ for any $z \in \{-1, 1\}$.*

Before we continue with the proof, notice the following immediate corollary of Lemma 1

**Corollary 12.1.** *For every constant $c > 0$, there is a distribution $D$ over a pair of convex functions $\{f(\mathbf{w}; 1), f(\mathbf{w}; -1)\}$, such that $f(\mathbf{w}; z)$ is a 1-Lipschitz convex function in $\mathbb{R}^2$ and, for every $c < \eta < 1$ denote $\mathbf{v}_{z,\eta} = -\eta \nabla f(0; z)$. Then the following holds:*

- *For every $z \in \{-1, 1\}$ we have that $f(\mathbf{v}_{z,\eta}; z) = f(\mathbf{v}_{-z,\eta}; z)$.*

- *$\|\mathbf{v}_{z,\eta} - \mathbf{v}_{-z,\eta}\| > \eta/4$*

- *$\|\mathbf{v}_{z,\eta}\| \leq \eta$ for any $z \in \{-1, 1\}$*

Indeed to derive Corollary 12.1 from Lemma 1, take a distribution that w.p. $1/2$ picks $f(\mathbf{w}; 1)$ from Lemma 1, and with probability $1/2$ picks $f(\mathbf{w}; -1)$. One can observe that the result holds.

**Proof of Lemma 1** Let us define $f(\mathbf{w}; \pm 1)$ as follows, denote $\mathbf{v}_1 = (1/4, 3/4)$, $\mathbf{v}_{-1} = \frac{3}{4} \cdot \mathbf{e}_2$ and let

$$f(\mathbf{w}; z) = \max(0, \mathbf{v}_z^\top \mathbf{w} + c\|\mathbf{v}_z\|^2)$$

It is easy to check that $\nabla f(0; 1) = \mathbf{v}_1$ and that $\nabla f(0; -1) = \mathbf{v}_{-1}$, and that $\|\mathbf{v}_1 - \mathbf{v}_{-1}\| \geq 1/4$.

Next, note that if $\eta > c$ then

$$f(\mathbf{v}_{1,\eta}; 1) = \max\left(0, (-\eta + c) \cdot \|\mathbf{v}_1\|^2\right) = 0 = \max\left(0, -\eta \mathbf{v}_1^\top \mathbf{v}_{-1} + c\|\mathbf{v}_{-1}\|^2\right) = f(\mathbf{v}_{-1,\eta}; 1)$$

Similarly, $f(\mathbf{v}_{-1,\eta}; -1) = 0 = f(\mathbf{v}_{1,\eta}; -1)$.

## C.1 Proof of Theorem 3

Theorem 3 is an immediate corollary of the following theorem:

**Theorem 13.** *For every $T$, there exists a distribution $D$ over $\mathbb{R}^d$ with $d = \Theta(T)$ such that if we run SGD with step size $1/T^2 < \eta < 1$, the following holds: for any regularizer $r$, w.p. at least $1/10$ over the sample $S$ there is $\mathbf{w}_r \in \mathcal{W}$ such that*

$$\begin{aligned} F_S(\mathbf{w}_r) &\leq F_S(\mathbf{w}_S) , \\ r(\mathbf{w}_r) &\leq r(\mathbf{w}_S) , \end{aligned}$$

*Moreover*

$$\|\mathbf{w}_r - \mathbf{w}_S\|_2^2 = \Theta(\eta^2 T) .$$

To see how Theorem 3 follows, Let $\mathbf{w}^*$ be the minimizer of $r(\mathbf{w})$ amongst all $\mathbf{w} \in \mathcal{W}$ with $F_S(\mathbf{w}) \leq F_S(\mathbf{w}_S)$ then by strong convexity

$$r(\mathbf{w}_S) \geq r(\mathbf{w}^*) + \frac{\lambda}{2}\|\mathbf{w}_S - \mathbf{w}^*\|^2.$$

Now if $\|\mathbf{w}_S - \mathbf{w}^*\| > \frac{1}{4} \cdot \|\mathbf{w}_r - \mathbf{w}_S\|$ we are done. If not, then

$$\|\mathbf{w}_S - \mathbf{w}^*\| \leq \frac{1}{4}\|\mathbf{w}_r - \mathbf{w}_S\| \leq \frac{1}{4}[\|\mathbf{w}_r - \mathbf{w}^*\| + \|\mathbf{w}_S - \mathbf{w}^*\|],$$

which leads to $\|\mathbf{w}_r - \mathbf{w}^*\| \geq \frac{3}{4} \cdot \|\mathbf{w}_r - \mathbf{w}_S\|$. Using this, we get by strong convexity:

$$r(\mathbf{w}_S) \geq r(\mathbf{w}_r) \geq r(\mathbf{w}^*) + \frac{\lambda}{2}\|\mathbf{w}_r - \mathbf{w}^*\|^2 \geq r(\mathbf{w}^*) + \frac{9\lambda}{32}\|\mathbf{w}_r - \mathbf{w}_S\|^2.$$

**Proof of Theorem 13** Choose $d = 10 \cdot T$ and let $\mathcal{W}$ be the unit ball of $\mathbb{R}^d$. Let $D_0$ be the distribution over convex functions in $\mathbb{R}^2$ whose existence follows from Corollary 12.1 with $c < 1/(4T^2)$.

We now define a distribution over convex functions in $\mathbb{R}^d$ as follows: at each iteration pick uniformly $\mathbf{z}$ from the set $\{\mathbf{z} = (z; i) : z \in \{-1, 1\}, i = 1, ..., 5T\}$ and let:

$$\mathbf{f}(\mathbf{w}; \mathbf{z}) = f((w_{2i-1}, w_{2i}); z).$$

To prove the result we proceed as follows: given a sample $S$ drawn i.i.d from the distribution $D$, let us call a sample point $\mathbf{z}_t = (z_t, i_t)$ *good* if $t < T/2$ and if $i_t$ appears only once in the sample (i.e. for any $t' \leq T$, $i_{t'} \neq i_t$). Denote by $S_g$ the set of good samples.

Next for a sample $S$ define a sample $S' = \{\mathbf{z}_1', \ldots, \mathbf{z}_T'\}$ to be a sample that differ from $S$ only at good sample points, and for every good sample point if $\mathbf{z}_t = (z_t, i_t)$ then $\mathbf{z}_t' = (z_t', i_t) = (-z_t, i_t)$. It is not hard to see that $S$ and $S'$ are identically distributed (though dependent).

Now first, we want to show that $F_S(\mathbf{w}_S) = F_S(\mathbf{w}_{S'})$ w.p. 1 and that w.p. 0.2 we have that

$$\|\mathbf{w}_S - \mathbf{w}_{S'}\| > \frac{\sqrt{T}\eta}{22}.$$

If we can show that, then we are done. Indeed because, by symmetry, we have with probability $1/2$ $r(\mathbf{w}_{S'}) \leq r(\mathbf{w}_S)$. We can then take $\mathbf{w}_r = \mathbf{w}_{S'}$. Taken together we have that with probability 0.1 all the requirements of the theorem hold.

**Claim.** $F_S(\mathbf{w}_S) = F_S(\mathbf{w}_{S'})$

*Proof.* Indeed, fix a sample $S$. To avoid cumbersome notations, and because $S$, $S'$ are fixed, we will denote here $\mathbf{w}_S = \bar{\mathbf{w}}$ and $\mathbf{w}_{S'} = \bar{\mathbf{w}}'$. Next, for a vector $\mathbf{w}$ and coordinate $i_t$ let us also denote $\mathbf{w}(i_t) = (w_{2i_t-1}, w_{2i_t}) \in \mathbb{R}^2$. Note that for any $\mathbf{w}$, and $\mathbf{z}_t$, the value $\mathbf{f}(\mathbf{w}; \mathbf{z}_t)$ depends only on $\mathbf{w}(i_t)$ (i.e. independent of the other coordinates). Also, for any $i$ and $t$, we have that $\mathbf{w}^{(t)}(i)$ depends only on $\mathbf{z}_{t'}$'s such that $t' \leq t$ and $i_{t'} = i$. In particular, for any $\mathbf{z}_t \notin S_g$ we have that $\bar{\mathbf{w}}(i_t) = \bar{\mathbf{w}}'(i_t)$, hence

$$f(\bar{\mathbf{w}}; z_t) = f(\bar{\mathbf{w}}(i_t); z_t) = f(\bar{\mathbf{w}}'(i_t); z_t) = f(\bar{\mathbf{w}}'; \mathbf{z}_t).$$

Next, we want to show that for a good coordinate $\mathbf{z}_t$ we also have that $f(\bar{\mathbf{w}}; z_t) = f(\bar{\mathbf{w}}'; z_t)$. For this, as in Corollary 12.1 let us denote for any $\eta$ and $z$ by $\mathbf{v}_{z,\eta} = -\eta \nabla f(\mathbf{0}; z) \in \mathbb{R}^2$. Then, for any good coordinate we can show that

$$\bar{\mathbf{w}}(i_t) = -\frac{T-t}{T} \eta \nabla f(\mathbf{0}; z_t) = \mathbf{v}_{z_t, \eta'}, \tag{34}$$

$$\bar{\mathbf{w}}'(i_t) = -\frac{T-t}{T} \eta \nabla f(\mathbf{0}; -z_t) = \mathbf{v}_{-z_t, \eta'} \tag{35}$$

where $\eta' = \frac{T-t}{T}\eta > \frac{1}{2}\eta > c$. Indeed, recall that we chose $c = 1/(4T^2)$. Thus, from Corollary 12.1 we obtain that $f(\bar{\mathbf{w}}(i_t), z_t) = f(\bar{\mathbf{w}}'(i_t), z_t)$ and in particular

$$\mathbf{f}(\bar{\mathbf{w}}; \mathbf{z}_t) = \mathbf{f}(\bar{\mathbf{w}}'; \mathbf{z}_t).$$

$\square$

**Claim.** *w.p. at least* $0.2$ *we have that*

$$\|\mathbf{w}_S - \mathbf{w}_{S'}\| > \frac{\sqrt{T}\eta}{22},$$

*Proof.* Again we will use the notation $\bar{\mathbf{w}} = \mathbf{w}_S$ and $\bar{\mathbf{w}}' = \mathbf{w}_{S'}$. Note that by Corollary 12.1, as well as Eqs. (34) and (35) we have that

$$\|\bar{\mathbf{w}}(i_t) - \bar{\mathbf{w}}'(i_t)\| \geq \eta/8,$$

for any good sample point $\mathbf{z}_t$. Now:

$$
\begin{aligned}
\|\bar{\mathbf{w}}' - \bar{\mathbf{w}}\|^2 &= \sum_{i=1}^{5T} \|\bar{\mathbf{w}}(i) - \bar{\mathbf{w}}'(i)\|^2 \\
&\geq \sum_{i \in S_g} \|\bar{\mathbf{w}}(i) - \bar{\mathbf{w}}'(i)\|^2 \\
&\geq \frac{|S_g|\eta^2}{64}.
\end{aligned}
\tag{36}
$$

Thus, we only need to show that $\mathbb{E}[|S_g|] > \frac{T}{5}$. Indeed, since $|S_g| < T/2$, we obtain by Markov's inequality that with probability $0.25$, $|S_g| > T/7$

To show that $\mathbb{E}[|S_g|] > T/5$, for a sample $S$, let $S_b$ contain all coordinates that collided (i.e. $\mathbf{z}_t$ such that for some $\mathbf{z}_{t'}$ we have that $i_t = i'_t$).

In order to calculate $|S_b|$ define $\chi_{t,t'} = I(i_t = i'_t)$ for every $t, t' \in [T]$. Note that $Pr(\chi_{t,t'} = 1) = \frac{1}{10T}$ and since there are at most $T(T-1)/2$ such pairs we get $\mathbb{E}[|S_b|] \leq \sum_{t,t'} Pr(\chi_{t,t'} = 1) \leq (T-1)/20$. Note that any coordinate $i_t$ with $t < T/4$ that did not collide is a good coordinate, hence

$$
\begin{aligned}
\mathbb{E}[|S_g|] &\geq T/4 - \mathbb{E}[|S_b|] \\
&\geq T/5
\end{aligned}
$$

$\square$

## C.2 Proof of Theorem 4

Again, let $D_0$ be the distribution from Corollary 12.1 with $c > \frac{1}{kT^2}$, for some constant $k$ (to be determined later). We define a distribution $D$ over $\mathbb{R}^d$, where we let $d = 100T \cdot k$. as follows: pick $k$ r.v $\{z^{(1)}, ..., z^{(k)}\} \in \{-1, 1\}$ and $k$ coordinates $\{i^{(1)}, \ldots, i^{(k)}\} \in [d/2]$, set

$$\mathbf{f}(\mathbf{w}; \mathbf{z}) = \frac{1}{k} \sum_{\ell=1}^{k} f((w_{2i^{(\ell)}-1}, w_{2i^{(\ell)}}), z^{(\ell)}).$$

As before, for a given sample $S$, let us define $S_g$ to be the set of "good samples" as follows: a tuple $(\mathbf{z}_t, \ell)$ is said to be *good* if $t < T/2$ and $i_t^\ell$ did not collide. Namely, any other sampled coordinate, $i_{t'}^{(\ell')}$ with $\ell' \in [k]$ and $t' \in [T]$ we have that if $i_{t'}^{(\ell')} = i_t^{(\ell)}$, then $\ell' = \ell$ and $t' = t$.

Next, for every sample $S$ define

$$\mathcal{S}(S) = \{S' = (\mathbf{z}'_1, \ldots, \mathbf{z}'_T) : i_t'^{(\ell)} = i_t^{(\ell)} \forall t, \ell \text{ and } \forall (\mathbf{z}_t, \ell) \notin S_g \ z_t'^{(\ell)} = z_t^{(\ell)} \}.$$

In words, $\mathcal{S}(S)$ includes all samples where at a good coordinate $(\mathbf{z}_t, \ell)$, $z_t^\ell$ may flip.

One can show that if we randomly pick $S$ and then pick uniformly an elements from $S' \in \mathcal{S}(S)$ then $S$ and $S'$ are identically distributed. As a corollary, if we pick a sample $S$ then w.p. 0.5 we have that

$$|\{\mathbf{w}_{S'} : S' \in \mathcal{S}(S), \ r(\mathbf{w}_{S'}) \le r(\mathbf{w}_S)\}| \ge \frac{|\mathcal{S}(S)|}{2}.$$

We next wish to prove that

$$\{\mathbf{w}_{S'} : S' \in \mathcal{S}(S)\} \subseteq \{\mathbf{w} : F_S(\mathbf{w}) \le F_S(\mathbf{w}_S)\}. \tag{37}$$

Once we show that, the result will follow from the following lemma (which we deter its proof to the end of the section).

**Lemma 4.** *For every $S$, let $A \subseteq \mathcal{S}(S)$ be a set such that $|A| > \frac{|\mathcal{S}(S)|}{2}$, then $A$ is $(2T, 10^{-4}\eta\sqrt{T})$-statistically complex*

We proceed by showing that Eq. (37) holds. The proof is very similar to the analog case in Theorem 3, and we will use similar notations: fix $S' \in \mathcal{S}(S)$ and use the shorthand notation $\bar{\mathbf{w}}$ for $\mathbf{w}_S$ and $\bar{\mathbf{w}}'$ for $\mathbf{w}_{S'}$, we will also use $\mathbf{w}(i, \ell) = (w_{2i^{(\ell)}-1}, w_{2i^{(\ell)}}) \in \mathbb{R}^2$. Another notation we add, as in Corollary 12.1, is as follows: for $\eta$ and $z$, $\mathbf{v}_{z,\eta} = -\eta \nabla f(0; z) \in \mathbb{R}^2$.

Then for any sample $(\mathbf{z}_t, \ell) \in S_g$, in $S_g$, we can show that

$$\bar{\mathbf{w}}(i_t, \ell) = -\frac{T-t+1}{kT}\eta\nabla f(0; z_t) = \mathbf{v}_{z_t^{(\ell)}, \eta_t'}, \tag{38}$$

$$\bar{\mathbf{w}}'(i_t, \ell) = -\frac{T-t+1}{kT}\eta\nabla f(0; z_t) = \mathbf{v}_{z_t'^{(\ell)}, \eta_t'}, \tag{39}$$

Where $\eta' = \frac{T-t+1}{kT}\eta > c$. Next, for any coordinate $(\mathbf{z}_t, \ell) \notin S_g$ we can show that $\bar{\mathbf{w}}(i, \ell) = \mathbf{w}'(i, \ell)$, hence if $\mathbf{z}_t$ is such that $(i_t, \ell_t) = (i, \ell)$ we clearly have that $f(\bar{\mathbf{w}}, z_t^{(\ell)}) = f(\bar{\mathbf{w}}', z_t^{(\ell)})$. Now for $(\mathbf{z}_t, \ell) \in S_g$, from Corollary 12.1 we obtain that

$$\begin{aligned}
\mathbf{f}(\mathbf{w}, z_t^{(\ell)}) &= \frac{1}{k} \sum_{\ell=1}^{k} f(\bar{\mathbf{w}}(i_t, \ell); z_t^{(\ell)}) \\
&= \frac{1}{k} \sum_{\ell=1}^{k} f(\mathbf{w}_{\eta_t', z_t^{(\ell)}}; z_t^{(\ell)}) && \textit{Eq. (38)} \\
&= \frac{1}{k} \sum_{\ell=1}^{k} f(\mathbf{w}_{\eta_t', z_t'^{(\ell)}}; z_t^{(\ell)}) && \textit{Corollary 12.1} \\
&= \frac{1}{k} \sum_{\ell=1}^{k} f(\bar{\mathbf{w}}'(i_t, \ell); z_t^{(\ell)}) && \textit{Eq. (39)} \\
&= \mathbf{f}(\bar{\mathbf{w}}', z_t^{(\ell)})
\end{aligned}$$

**Proof of Lemma 4** Fix $S$. Similar to the argument in Theorem 3, we have that with probability $0.2$, that $|S_g| > T \cdot k/7$. We claim that if this event occurred, with correct choice of $k$, then every subset of size $|\mathcal{S}(S)|/2$ will be $(2T, 10^{-4}\eta\sqrt{T})$-statistically complex.

Indeed, let us index the coordinates of $\mathbb{R}^{|S_g|}$ by the elements of $S_g$. Then, for every element $\mathbf{w} \in \{\mathbf{w}_{S'} : S' \in \mathcal{S}\}$ we let $\mathbf{u}(\mathbf{w}) : \mathbb{R}^d \to \mathbb{R}^{|S_g|}$ be an affine projection such that: if $(\mathbf{z}_t, \ell) \in S_g$, then $\mathbf{u}(w)_{(\mathbf{z}_t,\ell)}$ satisfies the following:

$$\mathbf{u}(w)_{(\mathbf{z}_t,\ell)} = \begin{cases} \frac{1}{\sqrt{|S_g|}} & \mathbf{w}(i_t,\ell) = \mathbf{v}_{1,\eta'_t} \\ -\frac{1}{\sqrt{|S_g|}} & \mathbf{w}(i_t,\ell) = \mathbf{v}_{-1,\eta'_t} \end{cases}$$

It can be seen from Eq. (38) and Eq. (39) and Corollary 12.1 that $\|\mathbf{v}_{1,\eta'_t} - \mathbf{v}_{-1,\eta'_t}\| > \frac{T-t+1}{kT}\eta/4 > \frac{\eta}{12k}$, hence we can define $\mathbf{u}$ to be $g$-Lipschitz where

$$g = \frac{24k}{\eta\sqrt{T}}.$$

Combining this with Corollary 6.1, we get that there exists a distribution $D$ over 1-Lipschitz convex functions such that, given $m = |S_g|/6 > Tk/62$ elements from $D$, with probability $1/4$ there is $\mathbf{w} \in A$ such that

$$\frac{1}{m}\sum_{i=1}^{m} f(\mathbf{w}, z_t) = 0.$$

but,

$$\mathbb{E}_{\mathbf{z}\sim D} f(\mathbf{w}, z) > 3/(g \cdot 4) > 0.003\frac{\eta\sqrt{T}}{k}$$

Taking $k = 124$, we get that $A$ is $(2T, 10^{-4}\eta\sqrt{T})$-Statistically complex.

# D   Proof of Theorem 5

We begin the construction by the definition of the distribution $D$:

$$f(\mathbf{w}; z = 1) = \begin{cases} w_1 & \text{if } \mathbf{w} \in A \\ 0 & \text{else} \end{cases}, \qquad\qquad f(\mathbf{w}; z = 3) = w_2,$$

$$f(\mathbf{w}; z = 2) = \begin{cases} -w_1 & \text{if } \mathbf{w} \in A \\ 0 & \text{else} \end{cases}, \qquad\qquad f(\mathbf{w}; z = 4) = -w_2$$

Where $z \sim Uniform([1, 2, 3, 4])$ and

$$A = \{(w_1, w_2) : |w_1|, |w_2| \le \frac{1}{4}\}.$$

Note that by symmetry $F = E_z[f(\mathbf{w}, z)] = 0$, and indeed in expectation this is a convex function. For the proof we will define two "good" events, set $c = \eta\sqrt{T} = \Theta(1)$, and let:

$$E_1 : |w_2^S| > \frac{1}{4},$$

$$E_2(\beta) : |w_1^S| > \frac{\eta\sqrt{T}}{2} \cdot \beta$$

where we write $\mathbf{w}_S = (w_1^S, w_2^S)$, and $\beta$ is a parameter sufficiently small so that.

$$\text{erf}(\beta) \le \sqrt{\text{erf}\left(\frac{\sqrt{50}}{4c}\right) - \text{erf}\left(\frac{\sqrt{50}}{4c}\right)}.$$

Note that $\beta$ depends only on $c = \Theta(1)$.

Let us denote by $E(\beta) = E_1 \cap E_2(\beta)$, then we will rely on the following claim that lower bounds the probability of the event $E$. We deter the proof of the claim to the end of the section and continue with the proof:

**Claim 14.** *Let $E(\beta) = E_1 \cap E_2(\beta)$ and suppose that $z \sim D$ then, for our choice of $\beta$, and sufficiently large $T$*

$$P(E(\beta)) > 1 - \sqrt{erf\left(\frac{50}{4c}\right)}.$$

*Proof.* Taking Claim 14 into account, Fix a random sample $S$. Let $\beta$ and $T$ be as in Claim 14 and assume that event $E := E(\beta)$ occurred. Throughout, let us denote $c = \eta\sqrt{T}$.

To show that the statement holds, we define $\mathbf{w}_0^* = (0, \bar{w}_2^S)$ and $\mathbf{w}_{-1}^* = (-w_1^S, \bar{w}_2^S)$. We will show that for one of these candidate vectors the statement holds.

First we want to show that if $\eta = \Theta(1/\sqrt{T})$, then $\|\mathbf{w}_S - \mathbf{w}_0^*\| = \Theta(1)$. Indeed, note that since $E_2(\beta)$ occurred

$$\|\mathbf{w}_S - \mathbf{w}_0^*\|_2 \geq |w_1^S| \geq \frac{\eta\sqrt{T}}{2} \cdot \beta = \Theta(1).$$

Similarly $\|\mathbf{w}_S - \mathbf{w}_{-1}^*\| = \Theta(1)$.

Next we want to show that $F_S(\mathbf{w}_0^*) \leq F(\bar{\mathbf{w}})$, or $F_S(\mathbf{w}_{-1}^*) \leq F(\bar{\mathbf{w}})$. Note that for every $\mathbf{w}$ such that $|w_2| \geq \frac{1}{4}$, for every $z = \{1, 2, 3, 4\}$, $f(\mathbf{w}; z)$ depends only on the second coordinate, namely $w_2$. In particular, if $|w_2^S| \geq \frac{1}{4}$ we obtain by the construction that $F_S(\mathbf{w}_S) = F_S(\mathbf{w}_0^*) = F_S(\mathbf{w}_{-1}^*)$. Thus, due to event $E_1$ we obtain the desired result.

Finally, we want to show $\min\{r(\mathbf{w}_0^*), r(\mathbf{w}_{-1}^*)\} < r(\mathbf{w}_S)$, w.p probability at least $1/4$. First, assume that with probability $1/2$ we have that $r(\mathbf{w}_{-1}^*) \neq r(\mathbf{w}_S)$. By symmetry one can show that in this case we have that $r(\mathbf{w}_{-1}^*) < r(\mathbf{w}_S)$ with probability $1/2$. Next, assume that $r(\mathbf{w}_{-1}^*) = r(\mathbf{w}_S)$ with probability at least $1/2$. In this case, we obtain that:

$$\begin{aligned} r(\mathbf{w}_0^*) &= r(0.5 \cdot \mathbf{w}_S + 0.5 \cdot \mathbf{w}_{-1}^*) \\ &< \max(r(\mathbf{w}_S), r(\mathbf{w}_{-1}^*)) \\ &= r(\mathbf{w}_S) \end{aligned}$$

$\square$

We are left with proving Claim 14

**Proof of Claim 14**   We will bound each event $E_1, E_2$ separately. We begin by bounding the event $E_1$:

**Bounding $E_1$:**   For $E_1$ we claim the following:

$$Pr\left(|w_2^S| \leq \frac{1}{4}\right) \leq \text{erf}\left(\frac{\sqrt{50}}{4c}\right) + \sqrt{\frac{50^3}{T}} \tag{40}$$

where $\Phi$ is the CDF of a mean zero unit variate normally distributed random variable, and erf is the error function, namely $\text{erf}(x) = 1 - 2\Phi(-x)$.

Note that if $\eta = O(\frac{1}{\sqrt{T}})$, given the above bound, the probability that $|w_2^S| > \frac{1}{4}$ is a constant.

*Proof.* Recall that

$$w_2^S = \frac{1}{T}\sum \eta(T - t)\frac{\partial f(\mathbf{w}^{(t)}, z_t)}{\partial w_2},$$

and one can observe that $\frac{\partial f(\mathbf{w}^{(t)}, z_t)}{\partial w_2}$ equals 1 w.p. $1/4$, $-1$, w.p $1/4$ and 0 w.p $1/2$, independently of $z_{t'}$ for $t' \neq t$.

Hence, applying Lemma 2, with $c = \eta\sqrt{T}$, $k = 1$ and $a = \frac{\sqrt{50}}{4c}$ we obtain that

$$\begin{aligned} P(-\frac{1}{4} \leq w_2^S \leq \frac{1}{4}) &= P\left(-\frac{\sqrt{50}}{4c}\frac{c}{\sqrt{50}} \leq w_2^S \leq \frac{\sqrt{50}}{4c} \cdot \frac{c}{\sqrt{50}}\right) \\ &\leq \text{erf}\left(\frac{\sqrt{50}}{4c}\right) + \sqrt{\frac{50^3}{T}} \tag{41} \end{aligned}$$

$\square$

We next move on to bound $E_2$

**Bounding $E_2$:** Let us consider a random sample $S' = \{z'_1, \ldots, z'_T\}$ that is generated by picking a random sample $S = z_1, \ldots, z_T$ i.i.d distributed according to $D$, and then for every $z_t$ such that $z_t \in \{1, 2\}$ with probability half we let $z'_t = 1$ and with probability half we let $z'_t = 2$. It can be seen that $S'$ is an i.i.d sequence drawn according to the distribution $D$.

Next, let us denote $c = \eta\sqrt{T}$, and a parameter $\alpha$ (to be chosen later). Define the event

$$E_\tau : \{S : \min\{t : \mathbf{w}^{(t)} \notin A\} > \frac{T}{\alpha \cdot c}\}.$$

For our choice of $\beta > 0$ we claim that for every $\alpha > 0$

$$Pr\left(|w_1^{S'}| < \frac{\beta}{\sqrt{50\alpha}}\Big| S, S' \in E_\tau\right) \leq 2\mathrm{erf}(\beta) + 2\sqrt{\frac{50^2 c^3 \alpha}{T}}$$

Indeed, Given $S$, let $\tau = \min\{t : \mathbf{w}^{(t+1)} \notin A\}$ and set $S'_\tau = \{z'_1, \ldots, z'_\tau\}$ and denote

$$X_\tau = \frac{1}{\sqrt{T}}\sum_{t=1}^\tau c\frac{T-t}{T}x_t.$$

where $x_t$ are i.i.d random variables such that w.p. $1/4$ equals 1, w.p. $1/4$ equals $-1$ and w.p. $1/2$ equals 0. Due to symmetry we have that:

$$Pr\left(|w_1^{S'}| < \frac{\beta}{\sqrt{\alpha \cdot c}}\Big| S, S' \in E_\tau\right) \leq 2Pr\left(|X_\tau| < \frac{\beta}{\sqrt{\alpha \cdot c}}\Big| S, S' \in E_\tau\right)$$

One can observe that

$$X_\tau = \sum_{t \in I}\eta\frac{T-t}{T}x_t = \frac{1}{\sqrt{T}}\sum_{t \in I}c\frac{T-t}{T}x_t.$$

Thus applying again Lemma 2 with $c = \sqrt{T}\eta$, $I = \{1, \ldots, T/k\}$ with, $k = \alpha \cdot c^3/50$ and $a = \beta$, we obtain the desired result.

Next, we want to bound $P(E_\tau)$. Now assume that for some $t < T/(\alpha \cdot c)$, we have that $\mathbf{w}^{(t)} \notin A$.

Let $T_\alpha = T/(\alpha \cdot c)$ and let $Z_1, \ldots, Z_{T_\alpha}$, be i.i.d copies of a random variable such that $P(Z_t = 1) = P(Z_t = -1) = 1/2$. Then

$$P(\neg E_\tau) \leq 4P\left(\min\{t : \eta\sum_{i=1}^t Z_i > \frac{1}{4}\} < \frac{T}{\alpha \cdot c}\right)$$

$$\leq 4P\left(\min\{t : \eta\sum_{i=1}^t Z_i > \frac{1}{4}\} < T_\alpha, \eta\sum_{i=1}^{T_\alpha} Z_i \geq \frac{1}{4}\right) + 4P\left(\min\{t : \eta\sum_{i=1}^t Z_i > \frac{1}{4}\} < T_\alpha, \eta\sum_{i=1}^{T_\alpha} Z_i \leq \frac{1}{4}\right)$$

$$= 8P\left(\eta\sum_{i=1}^{T_\alpha} Z_i \geq \frac{1}{4}\right) \tag{42}$$

where the last inequality is by symmetry (reflection principle). Next, by applying Hoeffding's inequality we obtain that

$$P(\eta\sum_{t=1}^{T_\alpha} Z_t \geq \frac{1}{4}) = P(\frac{\alpha\eta}{\sqrt{T}}\sum_{t=1}^{T_\alpha} Z_t \geq \frac{\alpha}{4\sqrt{T}}) = P(\frac{\alpha c}{T}\sum_{t=1}^{T_\alpha} Z_t \geq \frac{\alpha}{4\sqrt{T}}) \leq e^{-\frac{\alpha c}{32}}$$

Taken together we obtain that

$$P(\neg E_\tau) \leq 8e^{-\frac{\alpha c}{32}},$$

and

$$P(\neg E_2(\beta)) \leq P(\neg E_2|E_\tau)P(E_\tau) + P(\neg E_\tau)$$

$$\leq \mathop{\mathbb{E}}_S\left[P(|w_1^{S'}| < \frac{\beta}{\sqrt{\alpha c}}|S, S' \in E_\tau)\right] + P(\neg E_2)$$

$$\leq 2\mathrm{erf}(\beta) + 2\sqrt{\frac{50^2 c^3 \alpha}{T}} + 8e^{-\frac{\alpha c}{32}} \tag{43}$$

which yields the desired result.

**Bounding $E(\beta)$:**   Eqs. (41) and (43) yields then:

$$P(\neg E) < P(\neg E_1) + P(\neg E_2(\beta))$$
$$\leq \mathrm{erf}\left(\frac{\sqrt{50}}{4c}\right) + 2\mathrm{erf}(\beta) + 3\sqrt{\frac{50^2(50 + c^3\alpha)}{T}} + 8e^{-\frac{\alpha \cdot c}{32}}$$

Choosing $\beta$ sufficiently small, one can see that for large enough $\alpha$ and $T$ we obtain the desired result.