[Reviews · NeurIPS 2020]

Review 1

Summary and Contributions: This paper studies the implicit bias of SGD, and show that for a fairly large class of learning problems, the SGD outputs a solution that can not be a solution of a regularized problem.

Strengths: This paper provides a very interesting results on the implicit bias of algorithm. Instead of studying one specific scenario of implicit bias, the author study whether implicit bias exists for a very general class of learning problems (even though in the convex setting this is highly non-trivial). The negative results in the paper provide a very interesting direction to the community that one should seek other alternatives as explaining the generalization property of SGD.

Weaknesses: I enjoy reading this paper a lot. I do not see great limitations here.

Correctness: The paper is technically correct.

Clarity: Smooth and well written.

Relation to Prior Work: Yes the author clearly discuss the relation with previous works.

Reproducibility: Yes

Additional Feedback:


Review 2

Summary and Contributions: The authors study the limitations of implicit regularization in the general setting of stochastic convex optimization (SCO) and in particular, study the question of whether generalization of SGD in the SCO setting can be explained ONLY via implicit regularization and not some other mechanisms. The primary contribution is demonstrating that, in certain regimes, the (averaged) output of SGD is not a solution to a regularized empirical risk minimization problem for a large class of regularizers. The results of this paper establish a form of separation between explicit and implicit regularization.

Strengths: Implicit regularization is an active area of reserach that has been attracting increasingly more attention in the machine learning community over the past years. The results are novel and interesting and this is the first work to study implicit regularization in the setting of stochastic convex optimization. Given the recent NeurIPS publications on implicit regularization, the general theme that this work is trying to address is pertinent to the NeurIPS community.

Weaknesses: While the investigated questions are interesting and important, I have some concerns regarding the correctness of the proofs as well as several claims made by the authors throughout the text. See my answers below for details.

Correctness: 1) The problem domain W is a closed and bounded subset of R^{d} (lines 126-127), yet the definition of SGD output (Eq. (2)) as well as GD output (Eq. (4)) does not project onto W. The domain of regularizer is also assumed to be W (see line 172). If definitions (2) and (4) are not a typo, this raises important issues regarding correctness of the proofs. See the point below. 2) In Line 191, strongly convex regularizers are also assumed to be 1-Lipshitz. This is impossible on unbounded domains (i.e., if SGD iterates or output is not projected onto W). I have not checked the proofs in detail, but here is an example where a proof step appears to be incorrect. In the extended version of the paper, appendix B.1, the equation following (33) applies strong convexity to the SGD output w_S defined in (2) without proving that it is inside the set W. 3) Please explain in what sense does Theorem 1 imply that generalization of SGD cannot be explained via implicit bias point of view. A concrete example would be helpful, as the it is unclear why Theorem 1 would rule out implicit bias point of view towards explaining generalization. I.e., even if algorithm does not return a minimum norm solution, maybe it is biased towards solutions with small enough norm. A concrete, formal example following Theorem 1 would be very helpful. 4) Note that implicit bias of iterative algorithms is ITERATION DEPENDENT. Hence, even if optimization path of SGD exactly coicided with regularization path of ridge regression, you could prove statements such as the one in Theorem 1 (for any fixed regularizer lambda*||w||_{2}^{2}, after enough iterations, SGD output will incur a larger regularization penalty that that of corresponding ridge regression solution). 5) The assumption r(0) = 0 is not as mild as it seems, since SGD is naturally biased towards points with small training loss. I.e., intially, very large and crude steps will be taken to go far away from 0, which can be seen as r(0) being large.

Clarity: The writing could be significantly improved. Below I list some examples that interferred with the reading. The Euclidean norm is undefined. Lines 116-117: Explain what you mean by the phrase "perhaps in the strictest sense". What is this "sense"? How does it relate to your work? Is it the strictest sense or not? Line 121: Explain what you mean by "unstable convex problem". An algorithm can be unstable, how can a problem be unstable? In what sense are the problems considered in the submitted paper are stable? Line 181: Explain what you mean by "almost wlog". In my opinion, quite a lot of generality is lost by the assumption r(0) = 0 (see the section on correctness above). Line 181: Phrases like "perhaps somewhat stronger" are unhelpful. Is the assumption stronger or not? Line 190-191: A family of regularizers that is both strongly convex and Lipshitz is not a "natural family of regularizers". Strong convexity and lipshitzness can only hold on bounded domains, thus even ridge regression does not fall within such a setup. Line 213: I belive "comparable penalty" should be "the same" penalty, since the definitino (6) also requires the "training loss"/"empirical error" to be smaller than the one that is returned by the algorithm in question. Line 275: The constraint set "unit ball" is refered to as a "regularizer", whereas previously regularizer was a function denoted by r( ). Line 279: Unfinished sentence. There are several issues with the therorem statements. I will focus on Theorem 1 for concreteness. a) T = Omega(1/(lambda * eta)) is ambiguous. I believe you should replace = with >=, since otherwise the theorem statement is meaningless. b) Theorem statements are missing quantifiers. For instance, in Theorem 1, you should say whether the result holds for ANY step size eta in the given range or just for SOME particular eta in the given range. c) It seems that you use \Theta notation to denote NON-NEGATIVE functions, please mention that somewhere.

Relation to Prior Work: It is clear how this work differs from the prior literature, since it is the first one to study implicit regularization in the setting of stochastic convex optimization. However, the related literature section could be improved. First, implicit regularization dates back to much earlier works than the ones concerning neural networks (the first theoretical work on implicit regularization is due to Buhlmann and Yu 2003 "Boosting with the l2 loss"). Given that the work the authors present has little or nothing to do with neural networks, the related work section should be expanded to include other authors who developed the theory of implicit regularization for gradient descent over the past ~20 years. Also, it is worth noting that one of the central reasons motivating the sutudy of early stopping and implicit regularization is the computational efficiency of the method in comparison to the explicit regularization schemes. Second, the authors should elaborate in more details what is meant by "an attempt towrads separation between learnability and regularization" (line 110). In my opinion, such a separation is not established in the submitted work (see my questions above). Third, the paragraph concerning stability, in addition to the clarity concerns raised above, is missing some imporant works. For instance, Hardt, Recht and Singer 2015 "Train faster generalize better" and Chen, Jin and Yu 2018 "Stability and Convergence Trade-off of Iterative Optimization Algorithms" should both be cited and put into an appropriate context regarding your work. Finally, given that the submitted work is fundamentally trying to establish a form of separation between explicit and implicit regularization, other important results are missing. For instance, Ali, Kolter and Tibshirani 2019 "A Continuous-Time View of Early Stopping for Least Squares" paper shows that for linear models, gradient descent, along its optimization path, is closely related to the regularization path of ridge regression, through the lens of excess risk of obtained models. Other examples also exist, for instance, the works of Raskutti, Wainwright and Yu 2014 "Early stopping and non-parametric regression: An optimal data-dependent stopping rule" and Wei, Yand and Wainwright 2017 "Early stopping for kernel boosting algorithms: A general analysis with localized complexities" show that early stopped gradient descent can be analyzed via the same statistical tools as the correspondig explicitly regularized problems over Euclidean balls. My overall impression is that authors are unfamiliar with several important results in the implicit regularization literature, which in turn hinders the presentation and the ability to put the submitted work into an appropriate context. In addition to the above, could the authors please elaborate on two claims in the introduction: lines 20-23: could the authors please cite some references where neural networks are trained without any regularization (no early stopping, no dropout, no weight decay, etc.) and yet they achieve a remarkable performance? lines 28-29: could the authors please clarify on the exact result in the book [4], where it is shown that implicit regularization is IDENTICAL to explicit l2 regularization? The only results I know show that implicit regularization is SIMILAR to explicit l2 regularization (and identical in some regimes only. e.g., running gradient descent to convergence on underdetermined linear systems of equations with the quadratic loss is equivalent to the ridge regularization path in the limit lambda-> 0).

Reproducibility: Yes

Additional Feedback: ----- UPDATE AFTER READING THE AUTHORS' RESPONSE ----- I thank the authors for their thorough response, which I have read in detail, and, which has clarified several of my questions. I am, however, not convinced that the proofs can be fixed as easily as the authors claim, as it seems that there is loss of generality in using different radiuses for the constraint set of SGD iterates and that of the regularized solution. it is not enough that the iterates stay in a constant size ball (changing the radius of the ball will also change the optimal regularized solution). A direct analysis of projected SGD would be difficult, since the favourable expressions for unconstrained SGD iterates obtained in Theorem 8 would no longer apply. Also, I do not agree with the third point in the authors response, on that the authors "do not claim that generalization cannot be explained via implicit bias". Indeed, such a claim is made even in the abstract (lines 11-12), and multiple times throughout the text. Overall, I think that the paper is promising and interesting, but it needs to undergo a serious revision and another round of reviews that would ensure its correctness. ---------------------------------------------------------------------------------------------------- For convenience of the authors, all of my primary concerns raised above are numbered. Subject to the authors response, I am willing to increase the score, most importantly, if the authors address the correctness issues and also, if the authors are willing to improve the literature review and put the submitted work into a better context, in particular, with respect to the prior works investigating connections between explicit and implicit regularization. However, in its current presentation the paper looks closer to a draft rather than a finished work and the overall writing needs to be significantly improved before publication (see the section on clarity). Some typos: Line 162: R -> R^{d} Line 292: refers -> refer Line 343: attain -> attains


Review 3

Summary and Contributions: UPDATE: One of the reviewers points out that projection steps are not being taken. The authors mostly brush off the reviewer's concerns about projection. However, in our discussion we could not resolve the issues using the authors logic in the rebuttal. Therefore, I decided to lower my score. I find the topic to be very interesting and am looking forward to seeing a revised version of this submission. ================== Neyshabur, Tomioka, and Srebro argued in their paper "In Search of the Real Inductive Bias: On the Role of Implicit Regularization in Deep Learning" that the risk of networks trained by SGD did not obey the usual bias--variance tradeoff: SGD was seen to train larger and larger networks without overfitting. Their hypothesis was that SGD likely introduced some sort of implicit bias/regularization, much like it did in simple linear models. By studying this implicit bias, we might understand the generalization mystery behind deep learning. This work studies implicit bias/regularization, but does so in a setting that is often considered to be extremely well understood, namely stochastic convex optimization. In this setting, the authors show that there are distributions where SGDs behavior cannot be explained in terms of an implicit regularization, complicating the idea that implicit regularization surely explains generalization in the more complex setting of deep learning. The first result shows that distribution dependence is necessary in the sense that, for every distribution-independent regularizer, there is a distribution on instances where SGDs find solutions that are suboptimal in terms of both empirical risk and the regularization penalty. The second result looks at distribution-dependent regularizers and shows that there exists a data distribution, for which every (now distribution-dependent) regularizer cannot explain generalization. More formally, the level set defined by SGDs solution and the regularized loss, contains classifiers with small empirical risk and large risk. Both of these results are in the overparameterized setting. The authors also study convex learning problems with dimensionality smaller than the number of training points.

Strengths: Theoretical results are novel and may potentially have broad impact. Prior work has ruled out specific regularizers, while this work introduces claims about all possible regularizers. I think this work is very timely because the theory of deep learning faces a number of major hurdles that seem to be distracting us from essential lessons. By returning to a well-known stochastic convex optimization setting, the authors have made nice progress.

Weaknesses: The paper does not resolve the obstructions that it uncovers, although I do not see this as a serious problem.

Correctness: I am not aware of any serious issues with correctness. However, the statement of Theorem 4 and its use of Definition 1 seems to have some potential issues. Note that the notion of a "statistically complex" set implies the existence of a distribution D that satisfies some properties. By the order of quantifiers, though, this is a different distribution than the distribution D in the statement of the Theorem 4. Is that meant to be? In Definition 1, K looks like a nonrandom set, whereas in Theorem 4 it is a random set (i.e., measurable with respect to the sample). Indeed, the quantification going on in Theorem 4 is very informal and needs to be rewritten more formally, because the current quantification explain whether the quantification over the regularizer happens before or after the sample is chosen.

Clarity: Overall, I found the paper to be clearly written. I list a few places where clarity could be improved. It would be useful to see a short description of the key properties of the data distribution that allows one to obtain the result in Theorem 4. Too many details are deferred to the supplement. Theorem 4 has several issues (raised above). The statement could also be made more clear by specifying the relationship between d and T and what variables are being taken to infinity in the asymptotic expression \Theta(1): just d, or d and T at some rate?

Relation to Prior Work: To the best of my knowledge, the authors mention key relevant work and place their work in the context of others.

Reproducibility: Yes

Additional Feedback:


Review 4

Summary and Contributions: The paper provides very interesting negative results on the question of implicit regularization of SGD for the non-trivial problem of stochastic convex optimization.

Strengths: 1. The authors make a good choice of setting (SCO) that allows them to derive strong negative results, i.e. ruling out large classes of regularizers by showing Pareto-sub-optimality of the SGD solutions. 2. The paper is technically strong and makes substantial novel contributions. 3. The authors are careful not to overstate their results and limitations are made very clear. (I very much appreciate such intellectual honesty in a time where this can no longer be considered a given) 4. The constructions are quite general and it seems this opens up avenues for analysis of other models beyond SCO. 5. The paper is very well written and a pleasure to read.

Weaknesses: As almost every well-founded approach, one needs to simplify the problem considered in exchange for getting strong and meaningful results. One could wish for a setting that is closer to "deep ML practice".

Correctness: Yes.

Clarity: Yes, very much so.

Relation to Prior Work: Yes.

Reproducibility: Yes

Additional Feedback:

[Author Response · NeurIPS 2020]

We thank the reviewers for their detailed comments and insightful suggestions. Below we address some specific
comments; we apologize for conciseness due to space constraints. (We thank Reviewers 1 and 8 for supportive reviews!).

**Reviewer 3:**

We thank the reviewer for a detailed and thorough review. Again, we apologize for our brevity; the nature of this topic
warrants a much more careful and elaborate discussion, and it is unfortunate that we cannot do that here.

1. We were indeed inaccurate here: we will add an explicit projection step to the exposition of SGD and GD. As we
stress below, this does not affect our technical development in any crucial way.

2. As stated, there is indeed a formal projection. Note that in all our constructions the trajectory remains at a bounded
constant sized-ball. Thus, for a bounded and closed $W$, the projection step is avoided and does not affect our
constructions—we will highlight this in the proofs where needed.

3. Indeed, Thm. 1 does not imply that generalization cannot be explained via implicit bias (we do not make such a claim).
It only discusses strongly convex regularizers. As such, it definitely does not rule out the possibility of a bias towards
solutions with *small enough norm*. Thm. 2, on the other hand, shows that for any (admissible) regularizer we can
construct an instance where SGD converges to a point $w_*$ even though there exists another point $w_r$ that has the same
empirical error but (strictly) better regularization penalty. We follow here the intuition that if $r$ models the implicit
bias of SGD then, given two solutions with same empirical error, SGD needs to choose (approximately) the one with
smaller regularization penalty.

4. When comparing the output of SGD after enough iterations over the unregularized regression loss vs. ridge regression
solution with fixed $\lambda$: in this case the output of SGD and the output of ridge regression solution are incomparable (in
the Pareto-optimality sense). Namely, SGD will have a larger regularization penalty whereas ridge regression will
have a larger empirical error; as such, it does not prove Thm. 1. The theorem states the existence of a problem where
SGD converges to a solution that is not Pareto-efficient w.r.t. the empirical loss and $\ell_2$ norm (not even approximately).

5. The reviewer is correct here that this requires further explanation. First, note that the assumptions of admissibility are
used (and stated) only in Thm. 2, where we rule out *distribution-independent* regularizer. Any regularizer where
$r(0) \neq \min r(w)$ can be ruled out in this setting by simply considering the zero function (i.e., $f(w, z) = 0$ for all
$z$). In this case SGD converges to zero, which by assumption is $r$-suboptimal. Hence, we may assume here that
$r(0) = \min r(w)$ and we only need to normalize by choosing $\min r(w) = 0$. (again, we emphasize that this concerns
only distribution independent bias; in the distribution dependent section we are not making any assumption on the
regularizer to begin with). Following this discussion we will revise the assumption to $\min r(w) = 0$ (and not $r(0) = 0$)
and incorporate the discussion above in Thm. 2 where necessary.

Several other remarks:

• We thank the reviewer for indicating some important related work. The suggested lines of work are indeed relevant
and should be discussed. Thank you for pointing those out!

• Two works which we discuss that give evidence for training neural networks without regularizations are [14,22].
Specifically, [14] shows how training to zero training error with overcapacitated networks can improve test performance.

• The remark on where SGD is identical to introducing $\ell_2$ regularization: note that this claim is only for *linear*
optimization (hence not relevant for regression). Perhaps, a better reference for this fact is Shalev Shwartz "Online
Learning and Online Convex Optimization" (see Examples 2.3 therein).

**Reviewer 7:**

• *Typo in the notion of statistically complex set?*: No, the quantifiers are okay. Note that we want to understand if
the structure of the set $K$ can explain generalization, and without further considerations of the specific problem at
hand. For that one must choose a complexity measure that is independent of $D$. Note that any (successful) learning
algorithm converges, on a specific distribution, to a set of solutions that are okay for that distribution – thus changing
the quantifiers will lead to a tautology. What we desire here is to measure the complexity, or understand the structure
of the set of solutions w.r.t. any acceptable distribution. Specifically, Thm. 4 shows that on a given instance the set of
solution can be "too rich" in the sense that the capacity of the set cannot explain generalization.

• *Remarks on quantification and improving clarity of Thm. 4*: We will elaborate and clarify here. The quantification
over the regularizer is before the sample (i.e., the regularizer may depend on the distribution $D$ but is independent of
the sample $S$). We will also clarify the relationship between $d$ and $T$; in a nutshell, the asymptotic variable is $T$ (and
$d = O(T)$, $\eta = 1/\sqrt{T}$). Thanks for these suggestions!

[Meta-Review · NeurIPS 2020]

## Summary This paper investigates whether implicit bias can explain generalization in stochastic convex optimization (SCO). This responds to some recent work, which posited that generalization occurs because SGD possesses some implicit regularization property. By construction, the authors show that: (1) no distribution-independent implicit regularizer can be responsible for generalization; and (2) there are distributions for which no distribution-dependent implicit regularizer can explain generalization. Thus, implicit regularization can be ruled out for many instances of SCO. ## Strengths * The paper is well written and organized. The authors have clearly taken care to help the reader understand the results, supplying intuition and high-level takeaways of results as needed. * The results are relevant to contemporary discourse and will further our understanding of SCO and implicit regularization; they may change the direction of these discussions. ## Weaknesses * Understanding SCO may or may not bring us closer to understanding stochastic non-convex optimization, which is the more interesting setting in light of deep learning's success. Nonetheless, it is important to understand convex problems. * The paper's main contributions are negative results, with no recommendations for more promising directions to explain generalization. Perhaps that is a matter for future work, but it would have been nice if the results offered some advice. * The proofs are extraordinarily long and complicated, which makes them difficult to understand and verify. Perhaps with a few days (or weeks) of dedicated effort, one could thoroughly digest the proofs, but given the realistic constraints of life/work, most readers will just have to take the authors' word for it. (This will become important later on in my meta-review.) * Writing nit: there is no conclusion/discussion section, which makes the ending very abrupt. I guess there just wasn't enough space. I recommend shortening some of the background sections (2 & 3) to make room at the end for conclusions. ## Justification for Recommendation The only negative review came from R3, whose primary objection was that SGD and GD should be specified as _projected_ SGD and GD, respectively; otherwise, their iterates may end up outside the feasible set. The authors responded that this is a simple fix, which does not change their results, but R3 was not convinced. R3's argument goes as follows (paraphrasing the discussion): > In the setup, all models are assumed to belong to the unit ball, denoted W_1. Though, without projection, there is no guarantee that SGD or GD will keep the iterates in W_1. Th 1 claims that there exists a point inside W_1 that shows that SGD is Pareto-inefficient. The proof of Th 1 relies on Th 8, in which it is shown (p. 15 of supplemental) that the iterates are contained in a ball of radius 5 (W_5), not 1, and no projection is used. While over W_1, it was shown that the regularized ERM would have a lower regularization penalty and empirical error than that of the SGD solution, such a claim cannot be made over W_5; a regularized ERM solution over W_5 may have a larger regularization penalty than that of the W_1 solution; hence, the SGD solution may no longer be Pareto-suboptimal w.r.t. the regularized ERM over W_5. The proofs of all of these results are unfortunately very long and technical, and I must admit that I have not verified them in detail. Nonetheless, I believe that the issues R3 raises are not all that severe, and I will now attempt to explain why (using arguments borrowed from the discussion). First, it is clear that the iterates of SGD (as considered in the paper) never escape a constant-sized ball. In the construction used to prove Th 8, the radius is 5. So, if the feasible set is defined as a ball of radius 5, then the iterates of projected SGD will never require projection; they will naturally stay inside the feasible set. From this, we conclude that modifying the paper to explicitly state "projected SGD" is merely a technicality. However, since we have changed the problem domain from a 1-ball to a 5-ball, it is unclear whether the rest of the proof goes through with the 5-ball. To this I ask, can we not just change the construction used in Th 8 to ensure that the iterates never escape the 1-ball? 5 is not a magic number; surely some scaling of the construction (e.g., restricting \theta_1 to a different range) would result in the desired result. We are quibbling over constants that do not appear to be fundamentally important. The problem is, it will take me a very long time and much effort to verify this. So, lacking this effort, I am left with uncertainty. The thing is, even R3 isn't sure whether the results won't hold with the proposed fix; they have cast doubt on the proof, but have not provided their own proof of why the result is flawed. I will also note: as a result of the discussion, R7 decided to lower their score (from 7 to 5), though I believe that their original assessment was correct. (It is worth noting, the other 2 reviewers were not swayed to lower their scores.) I am thus going against the final scores and recommending acceptance. I think this is a good paper, and that it would be a shame to reject it due to a possible misunderstanding. ## Feedback for the Authors I _strongly_ encourage the authors to incorporate all of the feedback when revising the paper. In particular, I wish they could find a way to simplify the proofs in the appendix to make them more easily digestible. At the very least, adding more high-level intuitions (and maybe figures?), as they did in the main text, would go a long way.